# Secret-Protected Evolution for Differentially Private Synthetic Text Generation

**Tianze Wang**[1,2][*] **Zhaoyu Chen**[1][*] **Jian Du**[1][†] **Yingtai Xiao**[1]**, Linjun Zhang**[2]**, Qiang Yan**[1]
[1]TikTok, [2]Department of Statistics, Rutgers University

## Abstract

Text data has become extremely valuable on large language models (LLMs) and even lead to general artificial intelligence (AGI). A lot of high-quality text in the real world is private and cannot be freely used due to privacy concerns. Therefore, differentially private (DP) synthetic text generation has been proposed, aiming to produce high-utility synthetic data while protecting sensitive information. However, existing DP synthetic text generation imposes uniform guarantees that often overprotect non-sensitive content, resulting in substantial utility loss and computational overhead. Therefore, we propose Secret-Protected Evolution (SecPE), a novel framework that extends private evolution with secret-aware protection. Theoretically, we show that SecPE satisfies $(p, r)$-secret protection, constituting a relaxation of Gaussian DP that enables tighter utility–privacy trade-offs, while also substantially reducing computational complexity relative to baseline methods. Empirically, across the OpenReview, PubMed, and Yelp benchmarks, SecPE consistently achieves lower Fréchet Inception Distance (FID) and higher downstream task accuracy than GDP-based Aug-PE baselines, while requiring less noise to attain the same level of protection. Our results highlight that secret-aware guarantees can unlock more practical and effective privacy-preserving synthetic text generation.

## 1 Introduction

Text data has grown immensely valuable for large language models (LLMs), enabling these models to achieve revolutionary breakthroughs in natural language understanding and generation while delivering robust performance across document-understanding tasks—including classification, contextual autocompletion, and social recommendation (Chen et al., 2019; Mukherjee et al., 2020; Voytovich & Greenberg, 2022; Harte et al., 2023). However, training and adaptation typically rely on large volumes of private user text data, raising serious privacy risks including memorization and leakage of sensitive content (Carlini et al., 2019; 2021; Lukas et al., 2023; Wang et al., 2024).

To address privacy leakage, Differential Privacy (DP) (Dwork et al., 2006) has become the gold standard, offering a rigorous mathematical framework for mitigating information disclosure. Therefore, synthetic text based on DP can be safely shared and used for downstream tasks. A classical approach is to train a DP generator (Abadi et al., 2016) and then sample DP synthetic data (Yue et al., 2023; Yu et al., 2024; Tan et al., 2025). Despite its conceptual simplicity, such generators are computationally intensive, require hundreds of high-quality private data to achieve strong performance, and cannot directly leverage closed-source, state-of-the-art LLMs. More recently, *Private Evolution* (PE) (Lin et al., 2025) has emerged as an alternative: rather than privately training a model, one repeatedly queries a powerful foundation model to generate candidates, evaluates them against private data via DP voting, and resamples around the winners (Xie et al., 2024; Zou et al., 2025).

PE leverages strong off-the-shelf models and shifts the privacy cost to selection and aggregation. However, it still requires a substantial volume of private samples, and its pairwise similarity computations and iterative data processing make the pipeline highly inefficient. This inefficiency poses a critical challenge in practice and motivates the need for more scalable solutions. In addition, the

---

[1]indicates equal contributions. This work was done when Tianze Wang was an intern at TikTok.
[2]indicates corresponding author. E-mail: jian.du@tiktok.com

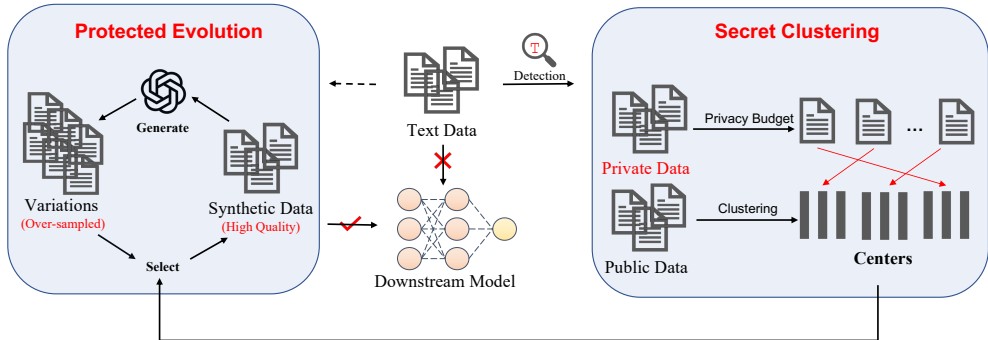

Figure 1: The overall of SecPE. The framework consists of two modules: (1) Secret Clustering: clustering is applied to public data and updated with noisy private data to form representative centers for voting; (2) Protected Evolution: in each iteration, candidate synthetic data consist of high-quality samples from the previous iteration together with their LLM-generated variations, and new high-quality samples are selected based on similarity to the noisy representatives.

reliance on canonical DP assumes that every record is equally sensitive, even though sensitive information may be sparse (Shi et al., 2022) and vary across users and attributes (e.g., medical records vs. movie ratings). Furthermore, secrets may repeat across records, and a single user may contribute multiple records; under user-level DP, this further degrades utility (Levy et al., 2021; Chua et al., 2024; Charles et al., 2024), increasing the noise required by uniform guarantees.

Recent work argues for *secret protection*, which provides guarantees tailored to specific secrets rather than membership (Ganesh et al., 2025). With predefined secrets, public data can be used without protection. For example, once the secrets are predefined, the corresponding sensitive information can be detected, allowing the remaining non-secret data to be freely used for clustering and summarization. This design enables the method to operate more efficiently, as the privacy budget is reserved exclusively for secret-related adjustments rather than being uniformly applied to the entire dataset. In the formulation, protection is calibrated at the adversary's prior for secrets and then directly bounding the reconstruction success probability, a similar concept as Hayes et al. (2023). Conceptually, this relaxes Gaussian DP (GDP) (Dong et al., 2019) by requiring protection only at a specific point $(p_j, r_j)$ on the trade-off curve, rather than over the entire curve. This relaxation weakens the privacy constraint but yields higher utility and more practical reconstruction protection. This perspective suggests re-thinking PE around *secret-aware* selection and aggregation rather than uniform DP noise.

In this paper, we introduce the **Secret-Protected Evolution (SecPE)** framework. As illustrated in Figure 1, SecPE consists of two key components: (1) *Secret Clustering*, which detects sensitive attributes and forms representative centers by updating public clusters with noisy private data; and (2) *Protected Evolution*, which iteratively samples variations from high-quality synthetic data, evaluates them against the noisy representatives, and selects the best candidates. This design preserves the practicality of PE while shifting protection toward secrets rather than uniform DP.

Our contributions are summarized as follows: (1) we propose a private synthetic data generation framework that emphasizes *secret protection* rather than canonical DP, thereby improving utility by reducing the noise typically required under DP; (2) we develop a secret-protected clustering method that substantially reduces runtime complexity compared to the PE approach, enabling scalability to larger datasets while maintaining competitive performance; and (3) through experiments on OpenReview, PubMed, and Yelp, we empirically demonstrate that SecPE achieves higher efficiency, lower Fréchet Inception Distance (FID) (Heusel et al., 2018), and better downstream accuracy than $\mu$-GDP–based PE baselines under the same reconstruction guarantees.

## 2 RELATED WORK

**Differential Privacy (DP).** We begin by reviewing $(\epsilon, \delta)$-DP and Gaussian Differential Privacy (GDP), the latter providing a clean bridge to secret-protection analysis.

**Definition 2.1** ($(\epsilon, \delta)$-DP). A randomized algorithm $\mathcal{M}$ is $(\epsilon, \delta)$-differentially private if, for any two neighboring datasets $D$ and $D'$ that differ in exactly one datapoint, and for all $\mathcal{S} \subset \text{Range}(\mathcal{M})$, it holds that $\Pr[\mathcal{M}(D) \in \mathcal{S}] \leq e^{\epsilon} \Pr[\mathcal{M}(D') \in \mathcal{S}] + \delta$.

Within the DP framework, an adversary aims to decide whether a specific record is present in a dataset. This naturally leads to a binary hypothesis test between two neighboring datasets $D \sim D'$. Following Dong et al. (2019), privacy can be characterized via the hypothesis-testing view through a trade-off function. Formally, let $P$ and $Q$ denote the output distributions of a randomized mechanism $\mathcal{M}$ on $D$ and $D'$, respectively. For any rejection rule testing $H_0 : P$ in favor of $H_1 : Q$, the trade-off function $T_{(P,Q)} : [0, 1] \to [0, 1]$ is defined as:

$$T_{(P,Q)}(\alpha) = \inf\{\beta_\phi : \alpha_\phi \leq \alpha\}, \qquad \alpha \in [0, 1], \tag{1}$$

where $\alpha_\phi = \mathbb{E}(\phi)$ and $\beta_\phi = 1 - \mathbb{E}(\phi)$ are type I and type II errors, respectively. Evidently, larger values of the trade-off function indicate a harder hypothesis testing problem (hence more private).

**Definition 2.2** (GDP). A randomized algorithm $\mathcal{M}$ satisfies $\mu$-Gaussian Differential Privacy if for every pair of neighboring datasets $D \sim D'$ with output distributions $(P, Q)$,

$$T_{(P,Q)}(\alpha) \geq G_\mu(\alpha) := \Phi\big(\Phi^{-1}(1 - \alpha) - \mu\big), \qquad \forall \alpha \in [0, 1]. \tag{2}$$

where $G_\mu(\alpha)$ is the benchmark trade-off curve for testing $\mathcal{N}(0, 1)$ against $\mathcal{N}(\mu, 1)$, and $\mu$-GDP asserts that distinguishing $\mathcal{M}(D)$ from $\mathcal{M}(D')$ is no easier than the Gaussian benchmark.

DP has its limitations: strong protection often entails utility loss, and its guarantees are typically uniform across all users and records, neglecting that not all secrets are equally sensitive. As a result, DP can be overly conservative for secret protection.

**DP Synthetic Text Generation.** The goal of generating DP synthetic texts is to mimic private data while protecting private information from leakage. An intuitive way is to train a language model with DP-SGD (Abadi et al., 2016) as a generator to guarantee DP for private data. While DP-Generator (Yue et al., 2023; Yu et al., 2024; Tan et al., 2025) is effective, it is computationally intensive and requires a large amount of high-quality private data to achieve strong performance. Furthermore, it cannot benefit from state-of-the-art LLMs, as these models (e.g., GPT, Claude, Gemini, etc.) are closed-source, which limits its potential. To handle the above issues, Private Evolution (PE) (Xie et al., 2024) is proposed for DP text synthesis, which only requires API access to foundation models and iteratively updates randomly initialized samples. Then, (Zou et al., 2025) and (Hou et al., 2025) further extend PE to data-deficient and data-isolated scenarios. However, standard DP provides uniform protection across all data, which can lead to excessive utility loss and computational overhead, especially when only a small portion of data is truly sensitive.

## 3 METHOD

To reduce the amount of noise added and thereby improve utility, we adopt secret protection in place of differential privacy. The formal definition of secret protection is provided in Section 3.1. Building on this definition, we introduce Secret-Protected Evolution (SecPE) in Section 3.2. In particular, Section 3.2.1 derives the noise scale required to achieve secret protection. Given this noise scale, Section 3.2.2 presents a private clustering method that satisfies the definition of secret protection, while significantly reducing computational complexity compared to Private Evolution. Finally, Section 3.2.3 summarizes the SecPE pipeline and establishes the privacy guarantee of the SecPE algorithm.

### 3.1 SECRET PROTECTION

The notion of secrets can be subjective, as it is determined by individuals' subjective will. On one hand, secrets may be proprietary business information that an organization seeks to protect (e.g., data containing trading details from an investment firm). On the other hand, secrets may comprise user information that a company wishes to leverage for model training. In general, secrets refer to sensitive content that warrants protection and the focus is not on membership privacy, but on safeguarding the secrets themselves against reconstruction. Ganesh et al. (2025) introduces a framework that provides privacy guarantees calibrated to the sensitivity of each secret, in contrast to DP, which enforces a uniform and often overly conservative level of protection.

**Definition 3.1** (Neighboring Datasets). Let $D = \{x_1, \ldots, x_n\}$ be a training dataset and $S = \{s_1, \ldots, s_m\}$ be a set of secrets. For each secret $s_j$, let $E_j(x_i)$ be an indicator function that equals 1 if $x_i$ contains secret $s_j$, and let $T_j(x_i, x_i')$ be an indicator that equals 1 if and only if $x_i$ and $x_i'$ differ only in the presence of secret $s_j$. We say that two datasets $D$ and $D'$ are *neighbors* with respect to secret $s_j$ (denoted $D \simeq_j D'$) if, for all $i \in [n]$, either $E_j(x_i) = 0$ or $T_j(x_i, x_i') = 1$.

Together, $E_j$ and $T_j$ specify which samples or datasets are considered neighbors under the secret-protection setting.

**Definition 3.2** (Secret Protection). Let $D = \{x_1, \ldots, x_n\}$ be a training dataset, where each sample may contain secrets from $S = \{s_1, \ldots, s_m\}$. For a secret $s_j \in S$, let $\pi_j$ denote a prior distribution over datasets $\{D_j^1, \ldots, D_j^K\}$ such that $\Pr(D_j^k) \leq p_j$, where $D$ and $D_j^k$ differ exclusively in the presence of $s_j$. A randomized mechanism $\mathcal{A}$ is said to satisfy $(\boldsymbol{p}, \boldsymbol{r})$-secret protection if, for any reconstruction attack $B$, the following holds:

$$\Pr_{D_j \sim \pi_j, \, \mathcal{A}} \left[ B\big(\mathcal{A}(D_j)\big) = s_j \right] \leq r_j, \quad \forall j. \tag{3}$$

Here, $\boldsymbol{p}$ and $\boldsymbol{r}$ are vectors, and $\pi_j$ encodes the adversary's prior knowledge about the secret $s_j$. The guarantee bounds, for each secret, the posterior reconstruction probability given a specified prior success probability.

This notion closely parallels the *reconstruction robustness* of Balle et al. (2022), where sharp bounds are obtained via the blow-up function. Accordingly, we interpret $(\boldsymbol{p}, \boldsymbol{r})$-secret protection through the lens of $\mu$-GDP, since the trade-off function is tightly connected to blow-up function. We align the neighboring relation so that $D \simeq D_j$ differ only by a single element: specifically, $s_j \in x \in D$ and $s_j \notin x' \in D_j$.

**Lemma 3.3.** *Any $\mu$-GDP mechanism $\mathcal{A}$ provides $(\boldsymbol{p}, \boldsymbol{r})$-secret protection, where*

$$r_j = 1 - \Phi\big(\Phi^{-1}(1 - p_j) - \mu\big). \tag{4}$$

*Proof.* Let $P$ and $Q_j$ denote the distributions of $\mathcal{A}(D)$ and $\mathcal{A}(D_j)$, respectively. By $\mu$-GDP, we have

$$\begin{aligned} T_{(P,Q_j)}(p_j) &\geq G_\mu(p_j) = \Phi\big(\Phi^{-1}(1 - p_j) - \mu\big) \\ \Longleftrightarrow 1 - B_{(P,Q_j)}(p_j) &\geq \Phi(\Phi^{-1}(1 - p_j) - \mu) \\ \Longleftrightarrow B_{(P,Q_j)}(p_j) &\leq 1 - \Phi(\Phi^{-1}(1 - p_j) - \mu) \triangleq r_j \end{aligned} \tag{5}$$

where $T_{(P,Q)}(\alpha) = \inf_{E:Q(E) \leq \alpha} P(E)$ is the trade-off function and $B_{(P,Q)}(\alpha) = \sup_{E:Q(E) \leq \alpha} P(E)$ is the blow-up function. By Theorem 2 of (Hayes et al., 2023), this directly implies $(\boldsymbol{p}, \boldsymbol{r})$-secret protection. $\square$

We use GDP to interpret secret protection because the posterior obtained from the trade-off curve yields a tight single-point bound at the specified prior. Note that the converse does not hold in general: $(\boldsymbol{p}, \boldsymbol{r})$-secret protection constrains the adversary's success at a single prior $p_j$, whereas $\mu$-GDP requires the entire trade-off curve $T_{(P,Q)}$ to lie above $G_\mu$. This highlights that $(\boldsymbol{p}, \boldsymbol{r})$-secret protection constitutes a relaxation of the classical DP definition.

### 3.2 SECRET-PROTECTED EVOLUTION

In this section, we propose **Sec**ret-**P**rotected **E**volution (SecPE), a framework that incorporates *secret protection* into the evolution paradigm introduced in prior work. Whereas traditional PE (Lin et al., 2025; Xie et al., 2024; Yu et al., 2024) operate under canonical DP guarantees, SecPE shifts the focus to *secret protection*, aligning with the discussion in the previous subsection.

Traditional PE draws random samples from a foundation model and iteratively refines candidates through DP voting on similarity to the private data. A key drawback of this approach is the high computational cost from redundant similarity evaluations. Moreover, the voting distribution is typically unbalanced, with a small subset of synthetic samples accumulating the majority of votes while the rest are selected almost uniformly (see Figure 4). This imbalance reveals inefficiencies in the selection procedure and motivates a more structured clustering-based design.

To address these issues, Algorithm 3 replaces direct voting with cluster representatives. At a high level, the SecPE pipeline consists of two stages: (1) *Secret Clustering*, where public data are clustered and updated with noisy private contributions to form representative anchors; and (2) *Protected Evolution*, where noisy representatives replace individual private samples in the voting process. This design preserves the practicality of PE while explicitly tailoring protection to secrets.

Specifically, for $M$ private examples and a target of $N_{\text{syn}}$ synthetic samples, the naive PE voting scheme requires $O(MN_{\text{syn}})$ similarity computations. In contrast, SecPE leverages Secret Clustering with $K$ anchors, reducing the complexity to $O(KN_{\text{syn}})$, where typically $K \ll M$. This yields substantial runtime savings in practice, as further validated in Table 2.

### 3.2.1 NOISE FOR SECRET PROTECTION

Following Ganesh et al. (2025), we assign a weight $w_i$ to each private example via linear program:

$$\max_{x_i \in D_{\text{pri}}} w_i \quad \text{subject to} \quad \sum_{x_i \in D_{\text{pri},j}} w_i \leq \Phi^{-1}(1-p_j) - \Phi^{-1}(1-r_j) \triangleq \eta_j, \quad w_i \in [0,1] \ \forall i. \quad (6)$$

where $D_{\text{pri},j} := \{ x_i \in D_{\text{pri}} \mid s_j \in x_i \}$ is a subset of $D_{\text{pri}}$ such that each data in $D_{\text{pri},j}$ contains secret $s_j$. The objective encourages including as many examples as possible; We then construct sampling probabilities $\rho_i = \frac{1}{\max_i w_i} \frac{w_i}{\sum_{i'} w_{i'}}$ to form a training subset. Here, $\eta_j = \mu$ in Equation 4 acts as a natural capacity constraint, permitting more samples to be selected when $s_j$ is less sensitive (i.e., larger $\eta_j$). However, secret protection only requires an *upper bound* on the blow-up function. In practice, $\eta_j$ may be chosen heuristically. A detailed procedure is outlined in Algorithm 1.

---

**Algorithm 1** Procedure SECRETNOISE

1: **Input:** Dataset $D_{\text{pri}}$, secrets $S_{\text{sec}}$, secret budget $(\boldsymbol{p}, \boldsymbol{r})$.
2: **Output:** noise parameter $\sigma$, sampling probabilities $\boldsymbol{\rho}$
3: $\{w_i\} \leftarrow$ solution to linear program (6) using the chosen $\eta_j$ values.
4: $V \leftarrow 1/\max_i w_i, \rho_i \leftarrow V \cdot \frac{w_i}{\sum_{i'} w_{i'}}$
5: For each $s_j \in S_{\text{sec}}, P_j = \mathcal{N}(\sum_{x_i \in D_{\text{sec},j}} \text{Bern}(\rho_i), \sigma^2)^{\otimes T}, Q_j = \mathcal{N}(0, \sigma^2)^{\otimes T}$
6: $\sigma_j \leftarrow \min\{\sigma : B_{(P_j,Q_j)}(p_j) \leq r_j\}, \quad \sigma \leftarrow \max_j \sigma_j$

---

### 3.2.2 SECRET CLUSTERING

In the PE procedure, a small number of synthetic samples receive the majority of votes, as illustrated in Figure 4. This phenomenon indicates that selection occurs in group-like clusters rather than being uniformly spread, which motivates replacing pointwise voting with representative voting via clustering. Predefined secrets allow us to first detect and cluster using only public data, and then apply a controlled shift informed by secret-containing data. In this way, the representative centers summarize the global structure of the dataset without directly exposing sensitive information. Synthetic data can then be selected based on proximity to these representative centers, eliminating the need to repeatedly process the entire dataset, a procedure that is especially costly for large-scale datasets.

A noteworthy hyperparameter is the number of clusters $K$. In practice, we recommend scaling $K$ with both the size of the original data and the target number of synthetic samples: (1) large enough to support diverse voting, and (2) small enough to limit noise amplification. Empirically, performance is largely insensitive to the exact choice of $K$ (see Section 4.2).

---

**Algorithm 2** Procedure SECRETCLUSTERING

---

1: **Input:** Dataset $D_{\mathrm{pri}} \cup D_{\mathrm{pub}}$, secrets $S_{\mathrm{sec}}$, text embedding model $\Psi$, clipping radius $R$.
2: **Input:** Public clusters $\{(e_k, n_k)\}_{k=1}^K = \mathrm{KMEANS}(D_{\mathrm{pub}}, K)$.
3: **Input:** $(\sigma, \boldsymbol{\rho}) = \mathrm{SECRETNOISE}(D_{\mathrm{pri}}, S_{\mathrm{sec}}, \boldsymbol{p}, \boldsymbol{r})$.
4: **Output:** Noisy cluster centers and cluster sizes $\{(\tilde{e}_k, \tilde{n}_k)\}_{k=1}^K$.
5: $E_{\mathrm{pri}} \leftarrow \mathrm{Clip}_R(\Psi(D_{\mathrm{pri}}))$; Initialize $e_k = n_k \cdot e_k, \ m_k = 0$
6: **for** $e_{\mathrm{pri},i} \in E_{\mathrm{pri}}$ **do**
7:     Sample $z \sim \mathrm{Bernoulli}(\rho_i)$.
8:     **if** $z = 1$ **then**
9:         Assign $e_{\mathrm{pri},i}$ to its nearest public center: $k \leftarrow \arg\min_{j \in [K]} d(e_{\mathrm{pri},i}, e_j)$.
10:         Update cluster statistics: $e_k \leftarrow e_k + e_{\mathrm{pri},i}, \ m_k \leftarrow m_k + 1$.
11:     **end if**
12: **end for**
13: $\tilde{n}_k \leftarrow n_k + m_k + \mathcal{N}(0, \sigma^2), \quad \tilde{e}_k \leftarrow \frac{e_k}{n_k + m_k} + \frac{2R}{n_k} \cdot \mathcal{N}(0, \sigma^2 \mathbb{I}_d)$.

---

**Theorem 1** (Secret Clustering). *Let $\{C_k\}_{k=1}^K \triangleq \{(\boldsymbol{e}_k, n_k)\}_{k=1}^K$ denote the set of public cluster centers with corresponding cluster sizes. Every private vector is clipped as $\hat{\boldsymbol{e}}_{\mathrm{pri},i} = Clip_R(\boldsymbol{e}_{\mathrm{pri},i}) = \boldsymbol{e}_{\mathrm{pri},i} \cdot \min\{1, R/\|\boldsymbol{e}_{\mathrm{pri},i}\|\}$, and then assigned to its nearest anchor; Let $m_k$ denote the number of private points assigned to anchor $\boldsymbol{e}_k$. For every cluster $k$, we release the perturbed statistics:*

$$
\tilde{\boldsymbol{e}}_k := \frac{n_k \cdot \boldsymbol{e}_k + \sum_{i \in C_k} \hat{\boldsymbol{e}}_{\mathrm{pri},i}}{n_k + m_k} + \xi_k, \quad \xi_k \sim \frac{2R}{n_k} \cdot \mathcal{N}(0, \sigma^2 I_d),
$$
$$
\tilde{n}_k = n_k + m_k + \eta_k, \qquad \eta_k \sim \mathcal{N}(0, \sigma^2). \tag{7}
$$

*where $\sigma$ is chosen by Algorithm 1 with $T = 1$. Then Algorithm 2 satisfies $(\boldsymbol{p}, \boldsymbol{r})$-secret protection.*

### 3.2.3 SECPE PIPELINE

Following Xie et al. (2024), Algorithm 3 instantiates SecPE with two interacting components each round: (i) Secret Clustering via Algorithm 2 to build noisy representatives and voting weights; and (ii) Protected Evolution that alternates selection with LLM-driven variation. Specifically, the procedure begins with an initialization step (line 4 in Algorithm 3) that prompts a foundation model to generate random samples. At each iteration, the top $N_{\mathrm{syn}}$ candidates from the previous round are selected based on their similarity to the secret-protected clustering centers. These survivors, together with their LLM-generated variations, form the candidate pool for the next round. A detailed convergence analysis is deferred to Appendix C.2.

---

**Algorithm 3** SecPE Pipeline

---

1: **Input:** Dataset $D_{\mathrm{pri}} \cup D_{\mathrm{pub}}$, secrets $S_{\mathrm{sec}}$, text embedding model $\Psi$
2: **Input:** Number of synthetic samples $N_{\mathrm{syn}}$, variation number $L$.
3: **Output:** Synthetic text dataset $S_{\mathrm{syn}}^T$
4: Initialize $S_0 \leftarrow \mathrm{RANDOM}(N_{\mathrm{syn}} * L)$
5: **for** $t \in \{0, 1, \cdots, T-1\}$ **do**
6:     $E_t \leftarrow \Psi(S_t), E_t \leftarrow E_t \cdot \min(1, R/\|E_t\|), \mathrm{Histogram}_t \leftarrow [0, \dots, 0]$
7:     $\{(\tilde{e}_k, \tilde{n}_k)\}_{k=1}^K \leftarrow \mathrm{SECRETCLUSTERING}(D_{\mathrm{pub}}, D_{\mathrm{pri}})$
8:     **for** $k \in \{1, \dots, K\}$ **do**
9:         $i \leftarrow \arg\min_{j:e_j \in E_t} d(\tilde{\boldsymbol{e}}_k, \boldsymbol{e}_j)$
10:         $\mathrm{Histogram}[i] \leftarrow \mathrm{Histogram}[i] + \tilde{n}_k$
11:     **end for**
12:     $S_{\mathrm{syn}}^{t+1} \leftarrow \mathrm{Top} \ N_{\mathrm{syn}}$ samples according to $\mathrm{Histogram}_t$.
13:     $S_{t+1} \leftarrow [\mathrm{VARIATION}(S_{\mathrm{syn}}^{t+1}, L), \ S_{\mathrm{syn}}^{t+1}]$
14: **end for**

---

**Theorem 2** (Privacy Guarantee for Algorithm 3). *Let Algorithm 3 run for $T$ iterations with noise multiplier $\sigma$ as specified in Algorithm 1. Then it satisfies $(\boldsymbol{p}, \boldsymbol{r})$-secret protection.*

Table 1: Hyperparameter settings for SecPE and Aug-PE across datasets.

| Dataset | $N_{syn}$ | Cluster $K$ | $L$ | Iterations | Temperature | Max tokens |
|---------|-----------|-------------|-----|------------|-------------|------------|
| OpenReview | 2000 | 15, 20, 15 | 6 | 10 | 1.2 | 448 |
| PubMed | 2000 | 2000, 3000, 4000 | 6 | 5 | 1.2 | 448 |
| Yelp | 5000 | 400, 600, 800 | 6 | 5 | 1.2 | 64 |

Table 2: LLM generation time and histogram computation time (seconds) for one epoch.

| Time (sec) | OpenReview | | PubMed | | Yelp | |
|------------|------|-----------|------|-----------|------|-----------|
| | LLM | Histogram | LLM | Histogram | LLM | Histogram |
| Aug-PE | 1698.7 | 126.9 | 828.5 | 32.2 | 347.1 | 30126.4 |
| SecPE | 1693.1 | **1.5** | 830.8 | **0.5** | 347.6 | **2.3** |

Theorem 2 provides a theoretical guarantee for Algorithm 3. The proof follows directly from Theorem 2.4 of (Doroshenko et al., 2022), since the pair of distributions $\mathcal{N}(\sum_{x_i \in D_{\mathrm{pri},j}} \mathrm{Bern}(\rho_i), \sigma^2)$ and $\mathcal{N}(0, \sigma^2)$ form a dominating pair in each round.

**Remark 3.4.** When cosine similarity is used as the distance metric, it suffices to apply $K$-means to $\ell_2$-normalized embeddings, which effectively transforms the clustering into cosine-based grouping. In this case, we set the sensitivity bound $R = 1$ when calibrating the noise scale.

## 4 EXPERIMENTS

### 4.1 SETUP

In this section, we empirically evaluate **SecPE** on text synthesis and privacy protection. We first consider a random word task to illustrate SecPE's ability to generate high-fidelity synthetic text under secret constraints. Then Personally Identifiable Information (PII) task that assesses protection of truly sensitive content, where PII is detected with AI4Privacy. In both tasks, SecPE delivers better utility at lower runtime: secret protection injects less noise than $\mu$-GDP at the same reconstruction budget, and representative voting with clustering cuts computation while preserving fidelity.

**Datasets.** We evaluate on three widely applied open-source datasets: (1) OpenReview (Xie et al., 2024): ICLR 2023 paper reviews labeled by research area and recommendation rating; (2) PubMed (Yue et al., 2023): medical paper abstracts; and (3) Yelp (Inc. Yelp, 2015): user reviews of businesses labeled by business category and rating.

**Baselines.** Given that *secret protection* is a new concept, we construct a baseline by instantiating the most popular Aug-PE under $\mu$-GDP, with $\mu$ set via Equation 4, and add Gaussian noise calibrated to the voting sensitivity following Xie et al. (2024). We use SecPE$_K$ to denote SecPE with $K$ clusters. Our comparison covers three aspects: (i) *Computational efficiency*: GPU hours for response generation and computation time for counting histogram are reported. (ii) *Downstream performance*: we fine-tune RoBERTa-base (Liu et al., 2019) on synthetic data to classify Yelp ratings/categories and OpenReview recommendations/areas. For PubMed, bert-base-uncased (BERT) Turc et al. (2019) is fine-tuned to report next-word prediction accuracy; (iii) *Real–synthetic similarity*: we compute FID on text embeddings and provide a comparison of text-length distributions;

**Implementation details.** As text generators, we use GPT-2 (Radford et al., 2019), Qwen-2.5-1.5B (Qwen et al., 2025) for main experiments. Llama-3.1-8B Dubey et al. (2024), Qwen-2.5-7B (Qwen et al., 2025), Mistral-7B-Instruct-v0.3 (Jiang et al., 2023) and GPT-4o-Mini (OpenAI, 2024) are applied for ablation study.

We use Sentence-Transformers (Reimers & Gurevych, 2019) as the embedding model $\Psi$. For the privacy budget, we fix the prior vector $\boldsymbol{p} = \mathbf{1} \cdot 10^{-4}$ and set the budget via the ratio $\boldsymbol{r}/\boldsymbol{p} = c$ with $c \in \{2, 10, 50, \infty\}$, where $c = \infty$ denotes the non-private setting. Although one could carefully tailor heterogeneous, secret-specific budgets to achieve better effectiveness, we adopt a uniform budget to enable a fair comparison with the $\mu$-GDP Aug-PE baseline, where $\mu = \Phi^{-1}(1-\boldsymbol{p}) - \Phi^{-1}(1-\boldsymbol{r})$.

Table 3: Performance comparison of downstream tasks within random words on PubMed.

| LLM | Method | $r/p = 2$ | | $r/p = 10$ | | $r/p = 50$ | | $r/p = \infty$ | |
|---|---|---|---|---|---|---|---|---|---|
| | | BERT_m | BERT_s | BERT_m | BERT_s | BERT_m | BERT_s | BERT_m | BERT_s |
| GPT2 | AugPE | 22.15 | 24.93 | 23.13 | 26.14 | 24.39 | 26.96 | **27.28** | **29.70** |
| | SecPE$_{2000}$ | 26.74 | 29.18 | 27.09 | 29.42 | 26.82 | 29.38 | 26.82 | 29.19 |
| | SecPE$_{3000}$ | **27.15** | 29.54 | **27.32** | **29.75** | 26.69 | 29.12 | 27.09 | 29.52 |
| | SecPE$_{4000}$ | 27.02 | **29.57** | 26.86 | 29.34 | **27.23** | **29.43** | 27.01 | 29.41 |
| Qwen-2.5-1.5B | AugPE | 20.37 | 22.65 | 21.01 | 23.09 | 21.18 | 23.52 | **23.68** | **25.87** |
| | SecPE$_{2000}$ | **23.17** | **25.37** | 22.26 | 24.40 | **22.93** | **24.96** | 22.50 | 24.63 |
| | SecPE$_{3000}$ | 22.31 | 24.48 | 22.24 | 24.55 | 22.65 | 24.92 | 22.84 | 24.91 |
| | SecPE$_{4000}$ | 22.17 | 24.41 | **22.63** | **24.99** | 22.84 | 24.91 | 22.29 | 24.40 |

For numerical stability, we approximate Mixture of Gaussian with a single Gaussian in both settings. Additional training and hyperparameter details are provided in Table 1. For each dataset, the hyperparameters are kept fixed across all methods. For generating LLMs' responses, the prompts are the same as (Xie et al., 2024).

## 4.2 PERFORMANCE COMPARISON ON RANDOM WORDS

In this task, we sort all vocabulary items in each dataset by frequency and designate words near the 20% quantile as secrets; a sample is treated as secret-containing if it includes any designated word.

**Runtime comparison.** Table 2 reports runtime on a NVIDIA A100 (80 GB) GPU. A key advantage of SecPE lies in its efficiency: *Secret Clustering* drastically reduces per-iteration histogram construction and selection time, accelerating the overall pipeline. Among Experiments, our method reduces this component by at least a factor of $60\times$; on Yelp (1.9M records), the reduction reaches roughly $10,000\times$. In terms of LLM sampling, SecPE achieves runtime comparable to Aug-PE, as both methods query the same model for the same number of variations.

**Memory and GPU Utilization.** Clustering before the PE step yields substantial resource savings. Concretely, in the Yelp experiment, we reduce memory usage by about 25.1 GB, since we no longer need to store embeddings for the entire dataset at once. GPU utilization also improves markedly: the original PE pipeline uses only about 3.2% of GPU capacity, whereas our approach reaches approximately 38.6%, indicating much more efficient hardware usage.

**Downstream Task.** Experimental results comparing SecPE with Aug-PE on downstream tasks are reported in Tables 3, 4 and 5. For each privacy budget and model, we highlight the highest classification accuracy in **bold**. On PubMed, as $r/p$ decreases from $\infty$ to 2, the BERT-small next-word prediction accuracy on Aug-PE (GPT-2) synthetic text drops from $29.70 \rightarrow 24.93$, whereas only a marginal change from $29.19 \rightarrow 29.18$ for SecPE$_{2000}$. The results show that, with the same number of training epochs, SecPE consistently achieves higher accuracy under private settings, and this advantage becomes more pronounced as the privacy requirement tightens (i.e., smaller $r/p$). In the non-private case ($r/p = \infty$), performance is slightly lower but broadly comparable, likely because clustering abstracts away fine-grained details and can occasionally induce mis-selections. We further observe non-systematic fluctuations in downstream accuracy when varying $K$, attributable to randomness in both the SecPE procedure and the downstream fine-tuning; overall, however, the method remains robust and not sensitive to the choice of $K$.

**Real–synthetic similarity.** The left two panels of Figure 2 align with the tabular results: as $r/p$ decreases, SecPE achieves lower FID (i.e., greater similarity to the original data) than Aug-PE, whereas in the non-private setting it yields higher FID. Moreover, FID varies little across all tested $K$, further indicating that the choice of $K$ does not materially affect performance. In the right two panels of Figure 2, we compare the empirical sequence-length distributions of synthetic data with those of the original corpus. As SecPE and Aug-PE rely on the same generative LLM, their length distributions are very similar.

**Ablation of LLM.** We evaluate SecPE$_{600}$ on Yelp under $r/p \in \{10, \infty\}$ using more advanced API-accessible LLMs. As shown in Table 6, these models achieve accuracy comparable to, or exceeding GPT-2 and Qwen-2.5-1.5B, indicating that our approach benefits from high-quality synthetic text

Table 4: Performance comparison of downstream tasks within random words on OpenReview.

| LLM | Method | $r/p = 2$ | | $r/p = 10$ | | $r/p = 50$ | | $r/p = \infty$ | |
|---|---|---|---|---|---|---|---|---|---|
| | | Area | Rating | Area | Rating | Area | Rating | Area | Rating |
| | Aug-PE | 29.06 | 25.70 | 27.94 | 27.12 | 32.48 | 27.88 | **41.06** | 28.70 |
| GPT2 | SecPE$_{15}$ | **30.77** | 30.26 | 31.88 | **30.70** | 30.34 | 28.27 | 39.02 | 28.38 |
| | SecPE$_{20}$ | 28.98 | **31.38** | 32.67 | 28.23 | 30.30 | 29.56 | 38.74 | **30.49** |
| | SecPE$_{25}$ | 30.34 | 29.24 | **34.81** | 30.66 | 32.48 | **30.31** | 38.60 | **30.49** |
| | Aug-PE | 32.70 | 25.55 | 32.23 | 25.80 | 36.49 | 28.52 | 40.20 | 28.09 |
| Qwen-2.5-1.5B | SecPE$_{15}$ | 38.34 | **27.73** | **38.67** | 26.02 | 36.09 | **30.95** | 36.03 | **32.03** |
| | SecPE$_{20}$ | 37.17 | 26.94 | 36.95 | 26.44 | 35.85 | 29.52 | 39.63 | 28.30 |
| | SecPE$_{25}$ | **38.92** | 27.66 | 37.03 | **27.82** | **40.24** | 28.81 | **40.24** | 28.86 |

Table 5: Performance comparison of downstream tasks within random words on Yelp.

| LLM | Method | $r/p = 2$ | | $r/p = 10$ | | $r/p = 50$ | | $r/p = \infty$ | |
|---|---|---|---|---|---|---|---|---|---|
| | | Category | Rating | Category | Rating | Category | Rating | Category | Rating |
| | Aug-PE | 71.53 | 47.02 | 71.62 | 54.72 | 72.60 | 64.02 | 73.54 | **65.28** |
| GPT2 | SecPE$_{400}$ | 72.06 | 61.44 | 72.90 | 58.92 | 72.18 | 64.30 | 72.50 | 61.28 |
| | SecPE$_{600}$ | 71.96 | 60.38 | 73.82 | 58.36 | **73.61** | **65.08** | 72.96 | 62.22 |
| | SecPE$_{800}$ | **72.74** | **62.46** | **74.28** | **63.70** | 73.12 | 63.82 | **73.58** | 62.48 |
| | Aug-PE | 72.70 | 55.84 | 72.14 | 53.52 | 71.93 | 55.54 | 72.14 | **59.02** |
| Qwen-2.5-1.5B | SecPE$_{400}$ | 73.97 | 55.80 | 72.00 | 57.80 | 73.28 | 57.70 | **73.84** | 58.78 |
| | SecPE$_{600}$ | 72.22 | 56.04 | **73.24** | 57.80 | **73.66** | 58.53 | 73.00 | 58.61 |
| | SecPE$_{800}$ | **74.12** | **58.64** | 72.60 | **58.66** | 72.54 | **60.93** | 73.14 | 57.06 |

produced by stronger LLMs. Within the same family, stronger variants tend to perform better. For example, at $r/p = 10$ (category, rating)-classification accuracy, GPT-4o-mini (74.84, 62.96) outperforms GPT2 (73.82, 58.36), and Qwen-2.5-7B (74.56, 63.06) outperforms Qwen-2.5-1.5B (73.12, 62.08). However, the 7B Mistral (72.52, 58.10) does not outperform a smaller LLM in our setting, suggesting that appropriate model selection, rather than parameter count alone, is crucial.

Table 6: Stronger LLM generators yield improved downstream accuracy on Yelp.

| LLM | $r/p = 10$ | | $r/p = \infty$ | |
|---|---|---|---|---|
| | Category | Rating | Category | Rating |
| GPT2 | 73.82 | 58.36 | 72.96 | 62.22 |
| Qwen-2.5-1.5B | 73.24 | 57.80 | 73.00 | 58.61 |
| Mistral-7B-Instruct-v0.3 | 72.52 | 58.10 | 73.38 | 61.28 |
| Llama-3.1-8B | 74.14 | 61.92 | 73.82 | 62.99 |
| Qwen-2.5-7B | 74.56 | **63.06** | 74.24 | **63.34** |
| GPT-4o-Mini | **74.84** | 62.96 | **75.10** | 63.28 |

**Ablation on the Number of Clusters** ($K$). Table 7 presents an ablation study over different choices of the number of clusters $K$, where SecPE is trained for 20 epochs. The results show that performance is largely insensitive to the exact choice of $K$, as long as $K$ is not too small, since an insufficient number of clusters fails to capture the overall data structure.

Table 7: Performance comparison on downstream tasks under different $K$ with $r/p = 10$.

| LLM | Label | $K=50$ | $K=100$ | $K=200$ | $K=800$ | $K=1200$ | $K=1600$ |
|---|---|---|---|---|---|---|---|
| GPT2 | Category | 62.50 | 63.88 | 70.94 | 73.76 | 73.56 | 73.92 |
| | Rating | 61.84 | 64.46 | 64.72 | 66.16 | 67.34 | 67.18 |

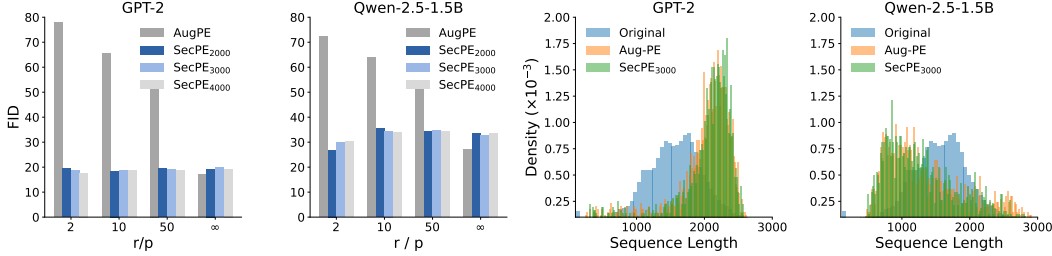

Figure 2: Results on PubMed. (Left) FID relative to the original data for SecPE and Aug-PE under $r/p \in \{2, 10, 50, \infty\}$ using GPT-2 and Qwen-2.5-1.5B. (Right) Synthetic sequence-length distributions for the non-private $\text{SecPE}_{3000}$ and Aug-PE generated by GPT-2 and Qwen-2.5-1.5B, compared with the original data.

## 4.3 Performance Comparison on Personally Identifiable Information (PII)

On the Yelp dataset, we detect 36 PII categories (e.g., age, email, gender) using AI4Privacy and Presidio and treat each category as a secret, yielding a dense secret-containing corpus. In this setting, the improvements over Aug-PE (see Table 8) are modest and less pronounced than in the random-word task. Crucially, the efficacy of our approach is intrinsically bottlenecked by the precision and recall of these underlying detection tools; since SecPE operates on the identified secrets, deploying more advanced and accurate PII detectors would naturally translate to more pronounced performance gains. Furthermore, it is worth noting that our comparison fixes the number of epochs across methods and therefore does not leverage SecPE's faster iteration speed.

Table 8: Performance comparison of downstream tasks within PII on Yelp.

| LLM | Method | $r/p = 2$ | | $r/p = 10$ | | $r/p = 50$ | | $r/p = \infty$ | |
|---|---|---|---|---|---|---|---|---|---|
| | | Category | Rating | Category | Rating | Category | Rating | Category | Rating |
| AI4Privacy | Aug-PE | 73.50 | 62.36 | 73.60 | **65.89** | 74.10 | **63.44** | **73.54** | **65.28** |
| | $\text{SecPE}_{600}$ | **73.65** | **63.45** | **75.05** | 61.22 | **74.34** | 62.45 | 72.96 | 62.22 |
| Presidio | Aug-PE | 71.72 | 55.80 | 72.54 | 63.34 | 74.15 | **66.92** | 75.14 | **67.04** |
| | $\text{SecPE}_{600}$ | **73.88** | **64.80** | **74.38** | **66.58** | **75.40** | 66.66 | **75.34** | 65.50 |

## 5 Conclusion

We introduced SecPE, a secret-aware evolution framework for privacy-preserving text synthesis. By calibrating protection at the level of secrets rather than enforcing uniform DP across all records, SecPE provides formal $(p, r)$-secret guarantees and relaxes Gaussian DP to the operative prior point, thereby achieving tighter utility–privacy trade-offs. Empirically, across diverse datasets, SecPE improves fidelity and downstream accuracy over GDP Aug-PE baselines under private settings, while also substantially accelerating the pipeline. Ablation studies further show that stronger LLMs consistently yield higher-quality synthetic text, highlighting the critical role of model selection. Overall, our results suggest that secret-aware mechanisms offer a more practical and effective approach to privacy-preserving text generation than DP, particularly in settings where sensitive content is sparse in type yet repeated across records and thus highly consequential.

## REPRODUCIBILITY STATEMENT

For the theoretical analysis of this work, we provide detailed explanations in Section 3.2. The assumptions and complete proofs of the theorems are presented in Appendix C. The datasets, models and hyperparameters used in the experiments are described in Section 4, and additional experimental settings are reported in Appendix B. All datasets used are publicly available, and their usage are explicitly referenced in the paper.

## ETHICS STATEMENT

This work adheres to the ICLR Code of Ethics, with no involvement of human subjects or animal experimentation. All datasets employed were sourced in compliance with relevant usage guidelines to ensure privacy protection. We ensured the avoidance of biases or discriminatory outcomes throughout the research process, utilized no personally identifiable information, and conducted no experiments posing privacy or security risks. Furthermore, we are committed to upholding transparency and integrity in the research. The proposed method facilitates privacy protection, and we aim to further standardize the use of private data.

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

## THE USE OF LARGE LANGUAGE MODELS (LLMS)

In this work, Large Language Models (LLMs) are used solely as general-purpose assistive tools to help polish and improve the clarity of the writing. Authors take full responsibility for all content in the paper, including text that was refined using LLMs, and confirm that no part of the manuscript generated by LLMs constitutes plagiarism or scientific misconduct.

## A    LIMITATION

While clustering accelerates the pipeline, it abstracts away fine-grained details, which can cause a modest loss of utility in the non-private regime. Another open challenge is the formal definition of what constitutes a secret and how to quantify its sensitivity. Future work will explore heterogeneous, secret-specific budgets and adaptive priors to further improve utility while maintaining protection. We also plan to extend SecPE to image domains (with an appropriate formalization of "secrets" in that setting) and investigate secret-protected generators.

## B    SUPPLEMENTARY EXPERIMENTS

### B.1    SIMULATION RESULT

To explicitly demonstrate how secret protection reduces the required noise relative to GDP, we consider a toy setup with $N = 8000$ records and $m = 400$ secrets. Each record contains each secret independently with probability Bernoulli$(0.01)$. For a fair comparison, $\mu$ in GDP is coupled to the $(\boldsymbol{p}, \boldsymbol{r})$ pair via $\mu = \Phi^{-1}(1-p) - \Phi^{-1}(1-r)$.

We report the noise ratio defined as $\sigma_{\text{GDP}}/\sigma_{\text{secret}}$ (larger is better). In the left panel, we fix $(N, m) = (8000, 400)$ and vary the privacy budget $r/p \in [2, 400]$. In the right panel, we fix $r/p = 10$ and $N = 8000$, and vary the number of secrets $m \in [100, 1000]$. Across both settings, the noise required by $(p, r)$-secret protection is consistently smaller than that of $\mu$-GDP.

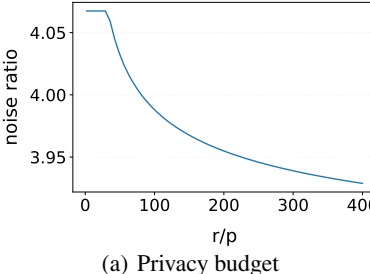
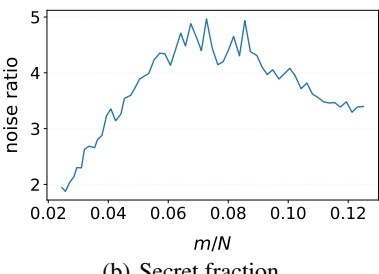

(a) Privacy budget         (b) Secret fraction

Figure 3: Noise ratio $\sigma_{\text{GDP}}/\sigma_{\text{secret}}$ comparing $(\boldsymbol{p}, \boldsymbol{r})$-secret protection with Gaussian DP. (a): $N = 8000$, $m = 400$, varying $r/p$. (b): $N = 8000$, $r/p = 10$, varying the number of secrets $m$.

### B.2    VOTING DETAILS

Here we compare raw voting in Aug-PE with post-clustering votes in SecPE. Figure 4 shows the first three labels on Yelp. The synthetic samples receiving the highest vote mass are largely the same across both methods, indicating that the clustering preserves the key selections and is thus reasonable.

### B.3    FID AND LENGTH

Figure 5 presents the FID with respect to the OpenReview dataset for SecPE and Aug-PE under $r/p \in \{2, 10, 50, \infty\}$, together with the synthetic sequence-length distributions of the non-private SecPE$_{20}$ compared against Aug-PE, using GPT-2 and Qwen-2.5-1.5B as generators.

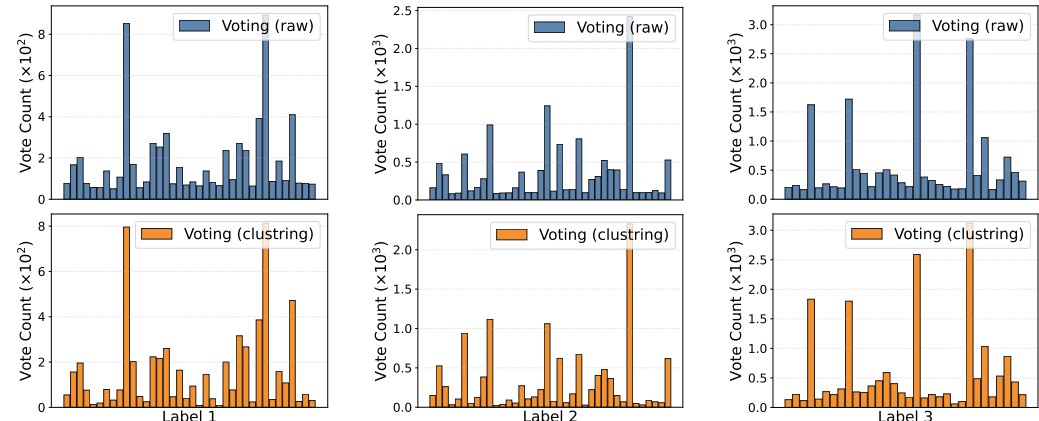

Figure 4: Voting distribution per label on Yelp. *Top:* raw votes from Aug-PE. *Bottom:* votes after clustering in SecPE.

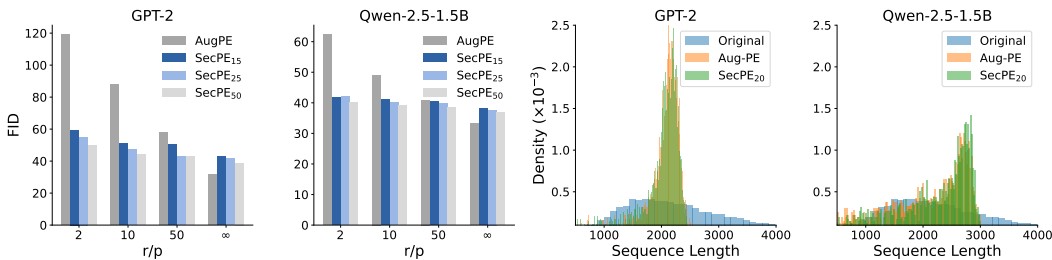

Figure 5: FID and sequence-length distributions on OpenReview.

Similarly, Figure 6 shows the FID with respect to the Yelp dataset for SecPE and Aug-PE under the same $r/p$ settings, along with the sequence-length distributions of the non-private $SecPE_{600}$ compared against Aug-PE, also using GPT-2 and Qwen-2.5-1.5B as generators.

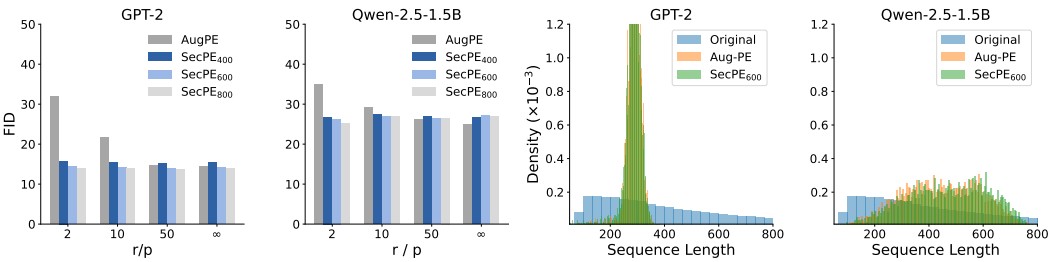

Figure 6: FID and sequence-length distributions on Yelp.

## B.4 APIs

Figure 7 shows the synthetic sequence-length distributions on Yelp for the non-private $SecPE_{600}$ across different generator models. Among them, Mistral-7B-Instruct-v0.3 exhibits the largest deviation from the original distribution, which aligns with its inferior performance reported in table 6.

## B.5 SUPPLEMENTARY PII EXPERIMENT

We conducted an additional PII experiment on the Yelp dataset using a stronger PII detection tool from Microsoft, Presidio. The new results in Table 9 show a clear and substantial improvement,

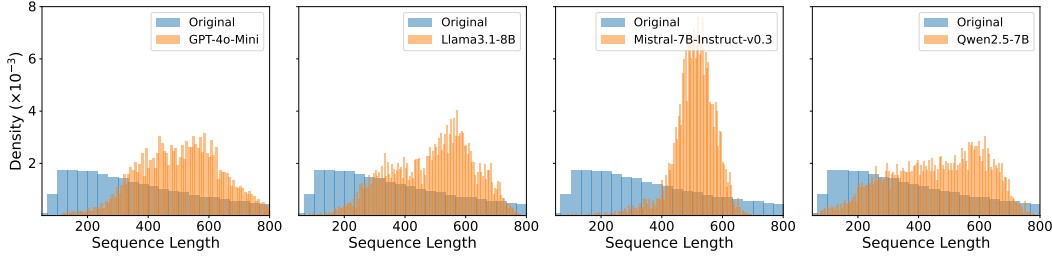

Figure 7: Synthetic sequence-length distributions across different generator models.

particularly under strong privacy settings (e.g., $r/p = 2$ or $r/p = 10$), confirming that SecPE remains effective when reliable secret detection is available.

Table 9: Performance comparison of downstream tasks within PII on Yelp.

| LLM | Method | $r/p = 2$ | | $r/p = 10$ | | $r/p = 50$ | | $r/p = \infty$ | |
|---|---|---|---|---|---|---|---|---|---|
| | | Category | Rating | Category | Rating | Category | Rating | Category | Rating |
| GPT2 | Aug-PE | 71.72 | 55.80 | 72.54 | 63.34 | 74.15 | **66.92** | 75.14 | **67.04** |
| | SecPE$_{600}$ | **73.88** | **64.80** | **74.38** | **66.58** | **75.40** | 66.66 | **75.34** | 65.50 |

## C ADDITIONAL THEORETICAL ANALYSIS

### C.1 FURTHER PROPERTIES AND RELATION TO DP

**Theorem 3** (Naive Composition). *Suppose $\mathcal{A}_1 : \mathcal{D} \mapsto \mathcal{R}_1$ satisfy $(\boldsymbol{p}_1, \boldsymbol{r}_1)$-secret protection, $\mathcal{A}_2 : \mathcal{D} \times \mathcal{R}_1 \mapsto \mathcal{R}_2$ satisfy $(\boldsymbol{p}_2, \boldsymbol{r}_2)$-secret protection. Define $\mathcal{A} : \mathcal{D} \to \mathcal{R}_1 \times \mathcal{R}_2$ by*

$$\mathcal{A}(D) = (\mathcal{A}_1(D), \mathcal{A}_2(D, \mathcal{A}_1(D)))$$

*Then $\mathcal{A}$ satisfies $(\boldsymbol{p}, \boldsymbol{r})$ secret protection such that, coordinate-wise,*

$$\boldsymbol{p} = \max(\boldsymbol{p}_1, \ \boldsymbol{p}_2), \quad \boldsymbol{r} = \boldsymbol{r}_1 + \boldsymbol{r}_2.$$

*Proof.* From definition, fix any secret $s_j$, to ensure that both mechanisms $\mathcal{A}_1$ and $\mathcal{A}_2$ satisfy their respective $(\boldsymbol{p}_1, \boldsymbol{r}_1)$ and $(\boldsymbol{p}_2, \boldsymbol{r}_2)$-secret protection guarantees, the prior distribution $\pi$ over $\{D_1, \ldots, D_K\}$ must satisfy:

$$\pi(i) \leq p_{1,j}, \quad \text{and} \quad \pi(i) \leq p_{2,j}, \quad \forall i \in [K]$$

Therefore, to apply both guarantees simultaneously, we define the composed mechanism's prior bound as:

$$p_j = \max(p_{1,j}, \ p_{2,j}),$$

so that any prior distribution satisfying $\pi(i) \leq p_j$ is valid for both mechanisms.

Let $B : \mathcal{R}_1 \times \mathcal{R}_2 \to [K]$ be any adversary attempting to identify the index $i$ from the output of $\mathcal{A}(D_i)$. Define an intermediate adversary $B_1(y_1) = \arg\max_i \Pr(\mathcal{A}_1(D_i) = y_1)$ (a hypothetical adversary trying to recover $i$ from $y_1$ only). By the assumption that $\mathcal{A}_1$ satisfies $(\boldsymbol{p}_1, \boldsymbol{r}_1)$-secret protection,

$$\Pr_{i \sim \pi, \ \mathcal{A}_1} [B_1(\mathcal{A}_1(D_i)) = i] \leq r_{1,j}$$

Since $\mathcal{A}_2$ satisfies $(\boldsymbol{p}_2, \boldsymbol{r}_2)$-secret protection, for each fixed $y_1$, the success probability of identifying $i$ from $\mathcal{A}_2(D_i, y_1)$ is at most $r_{2,j}$. Hence:

$$\Pr_{i \sim \pi, \ \mathcal{A}_2} [B(\mathcal{A}_2(D_i, y_1), y_1) = i \mid B_1(y_1) \neq i] \leq r_{2,j}$$

Thus, the total success probability of any adversary $B$ satisfies:

$$\Pr_{i \sim \pi, \ \mathcal{A}} [B(\mathcal{A}(D_i)) = i] \leq \Pr_{i \sim \pi, \ \mathcal{A}_1} [B_1(\mathcal{A}_1(D_i)) = i] + \Pr_{i \sim \pi, \ \mathcal{A}_2} [B(\mathcal{A}_2(D_i, y_1), y_1) = i \mid B_1(y_1) \neq i]$$

$$\leq r_{1,j} + r_{2,j}$$

Hence, the composed mechanism $\mathcal{A}$ satisfies $(\boldsymbol{p}, \boldsymbol{r})$ secret protection such that, coordinate-wise,

$$\boldsymbol{p} = \max(\boldsymbol{p}_1, \ \boldsymbol{p}_2), \quad \boldsymbol{r} = \boldsymbol{r}_1 + \boldsymbol{r}_2.$$

$\square$

**Remark C.1.** For every $\mu > 0$ the following equivalence holds Dong et al. (2019)

$$\text{the mechanism is } \mu\text{-GDP} \iff (\varepsilon, \delta(\varepsilon))\text{-DP for all } \varepsilon > 0,$$

where

$$\delta(\varepsilon) \ = \ \Phi\Big(-\frac{\varepsilon}{\mu} + \frac{\mu}{2}\Big) \ - \ e^{\varepsilon}\, \Phi\Big(-\frac{\varepsilon}{\mu} - \frac{\mu}{2}\Big).$$

Setting $\mu = \Phi^{-1}(1 - p_j) - \Phi^{-1}(1 - r_j)$ links the $(\epsilon, \delta)$-DP to $(p_j, r_j)$-secret protection.

Consider the same setting as Lemma 3.3: the neighboring datasets $D_1 \simeq D_2$ differ in exactly one element, specifically, $s_j \in x_1 \in D_1$ while $s_j \notin x_2 \in D_2$. The following lemma establishes a direct implication from $(\epsilon, \delta)$-DP to $(\boldsymbol{p}, \boldsymbol{r})$-secret protection.

**Lemma C.2.** *Any $(\epsilon, \delta)$-DP mechanism $\mathcal{A}$ provides at least $(\boldsymbol{p}, \boldsymbol{r})$-secret protection, where*

$$r_j = \frac{1}{1 + (e^{\varepsilon} + 1/c)^{-1} \cdot \frac{1 - p_j}{p_j}} + c \cdot \delta, \quad \forall c \geq 1$$

*Proof.* For any secret $s_j$, denote output distributions as:

$$P_1 := \mathcal{A}(D_1), \quad P_2 := \mathcal{A}(D_2)$$

and define the prior $\Pr[D_{\text{train}} = D_1] = p_j$, $\Pr[D_{\text{train}} = D_2] = 1 - p_j$. Let $y = \mathcal{A}(D_{\text{train}})$ denote an output from $\mathcal{A}$. By Bayes' rule, the posterior odds are:

$$\frac{\Pr[D_{\text{train}} = D_1 \mid y]}{\Pr[D_{\text{train}} = D_2 \mid y]} = \frac{\Pr(y \mid D_{\text{train}} = D_1)}{\Pr(y \mid D_{\text{train}} = D_2)} \cdot \frac{\Pr(D_{\text{train}} = D_1)}{\Pr(D_{\text{train}} = D2j)} = \frac{P_1(y)}{P_2(y)} \cdot \frac{p_j}{1 - p_j}$$

Now partition the output space $\mathcal{Y}$ into two parts:

$$G := \left\{ y \in \mathcal{Y} : \ \left| \log \frac{P_1(y)}{P_2(y)} \right| \leq \epsilon + t \right\}, \qquad U := \mathcal{Y} \setminus G$$

By the definition of $(\varepsilon, \delta)$-DP, we have:

$$e^{\varepsilon + t} P_2(U) < P_1(U) \leq e^{\epsilon} P_2(U) + \delta, \quad e^{\varepsilon + t} P_1(U) < P_2(U) \leq e^{\epsilon} P_1(U) + \delta$$

which implies $P_1(U) \leq \delta/(e^{\varepsilon + t} - e^{\varepsilon})$, $P_2(U) \leq \delta/(e^{\varepsilon + t} - e^{\varepsilon})$. For $c \geq 1$, let $t = \ln(1 + e^{-\varepsilon - \ln c})$ so that $P_1(U), \ P_2(U) \leq c\delta$. On the good region $G$, we have:

$$\frac{\Pr[D_{\text{train}} = D_1 \mid y]}{\Pr[D_{\text{train}} = D_2 \mid y]} \leq \left(e^{\varepsilon} + \frac{1}{c}\right) \cdot \frac{p_j}{1 - p_j} \Rightarrow \Pr[D_{\text{train}} = D_1 \mid y] \leq r^{\star} := \frac{1}{1 + (e^{\varepsilon} + 1/c)^{-1} \cdot \frac{1 - p_j}{p_j}}$$

Let $\mu$ denote the output distribution of $\mathcal{A}(I_{\text{train}})$,

$$\Pr[B(\mathcal{A}(D_i)) = i] \leq \int_{\mathcal{Y}} \max \left\{ \Pr[D_{\text{train}} = D_1 \mid y], \ \Pr[D_{\text{train}} = D_2 \mid y] \right\} d\mu(y)$$

We upper bound this by splitting the integral over $G$ and $U$:

$$\int_{\mathcal{Y}} \max(\cdot) \, d\mu = \int_G \max(\cdot) \, d\mu + \int_U \max(\cdot) \, d\mu$$
$$\leq r^{\star} \cdot \mu(G) + 1 \cdot \mu(U) \leq r^{\star} + c \cdot \delta$$

Thus, $\mathcal{A}$ provides $(p_j, \ r_j)$-secret protection in expectation, where

$$r_j = \frac{1}{1 + (e^{\varepsilon} + 1/c)^{-1} \cdot \frac{1 - p_j}{p_j}} + c \cdot \delta, \quad \forall c \geq 1.$$

For $\delta = 0$, letting $c \to \infty$, $(\epsilon, 0)$-DP implies $(\boldsymbol{p}, \boldsymbol{r})$-secret protection with

$$r_j \ = \ \frac{1}{1 + e^{-\epsilon} \frac{1 - p_j}{p_j}}.$$

$\square$

## C.2 Convergence Analysis

**Definition C.3** (Per-round mis-selection rate). The per-round mis-selection rate is the worst-case (over private points) probability that the selection event fails at round $t$:

$$\rho_t \triangleq \sup_{x \in D_{\text{priv}}} \Pr\big(\neg \text{Sel}_t(x) \,\big|\, S_t\big) \in [0, 1].$$

Here the probability is over the algorithmic randomness at round $t$ given $S_t$.

**Claim C.4.** *Fix a point $x \in D_{\text{pri}}$ and some iteration $t$. Suppose $z^* \in S_t$ is the closest point to $x$, $V = \text{VARIATION}(z^*, L) \cup \{z^*\}$. If $\|x - z^*\| \geq \eta$, then with probability at least $(1 - \rho_t)/2$, some point in $V$ will get noticeably closer to $x$ than $z^*$, i.e.,*

$$\min_{z \in V} \|x - z\|_2 \leq \left(1 - \frac{\log L}{4d}\right) \|x - z^*\|_2.$$

**Theorem 4.** *Assume that $\log L \ll d$. With probability $\geq 1 - \tau$, the non-private cluster-evolution algorithm outputs $S_{syn}$ with Wasserstein distance $W_p(D_{\text{pri}}, S_{syn}) \leq \eta$ after $T$ iterations. $\forall p \in [1, \infty]$ whenever*

$$T \gg \frac{4}{1 - \max_t \rho_t} \cdot \frac{d \log(D/\eta)}{\log L} + \log(N_{priv}/\tau) \quad \text{(or more generally, } \sum_{t=1}^{T}(1 - \rho_t) \gg \tfrac{4d \log(D/\eta)}{\log L}),$$
(8)

*Proof.* The proof of othertheorem C.4 and theorem 4 follows directly from Theorem 1 in Lin et al. (2025). In the idealized (no mis-selection) case, $T \gg 4\frac{d \log(D/\eta)}{\log L} + \log(N_{\text{priv}}/\tau)$. Accounting for per-round mis-selections at rate $\rho_t$ weakens the contraction by a factor $(1 - \rho_t)$, which replaces $T$ with $\sum_{t=1}^{T}(1 - \rho_t)$, or in the worst case, scales by $1/(1 - \max_t \rho_t)$. $\square$

**Theorem 5** (Secret Clustering). *Let $\{C_k\}_{k=1}^{K} \triangleq \{(e_k, n_k)\}_{k=1}^{K}$ denote the set of public cluster centers with corresponding cluster sizes. Every private vector is clipped as $\hat{e}_{\text{pri},i} = \text{Clip}_R(e_{\text{pri},i}) = e_{\text{pri},i} \cdot \min\{1, R/\|e_{\text{pri},i}\|\}$, and then assigned to its nearest anchor; Let $m_k$ denote the number of private points assigned to anchor $e_k$. For every cluster $k$, we release the perturbed statistics:*

$$\tilde{e}_k := \frac{n_k \cdot e_k + \sum_{i \in C_k} \hat{e}_{\text{pri},i}}{n_k + m_k} + \xi_k, \quad \xi_k \sim \tfrac{2R}{n_k} \cdot \mathcal{N}(0, \sigma^2 I_d),$$
$$\tilde{n}_k = n_k + m_k + \eta_k, \qquad \eta_k \sim \mathcal{N}(0, \sigma^2).$$
(9)

*where $\sigma$ is chosen by Algorithm 1 with $T = 1$. Then Algorithm 2 satisfies $(p, r)$-secret protection.*

*Proof.* Let $D_{\text{sec}}$ denote the dataset containing the secret associated with secret $s$, and let $D'_{\text{pri}} := D_{\text{pri}} \setminus D_{\text{sec}}$, where

$$D_{\text{sec}} := \{x \in D_{\text{pri}} \mid s \in x\}.$$

For any cluster $k$, suppose that $m_s$ is the number of private data containing $s$ that is assigned to cluster $k$, then the distribution of $m_s$ is $P := \mathcal{N}(0, \sigma^2)$ for $D'_{\text{pri}}$ and $Q := \mathcal{N}(\sum_{x_i \in D_{\text{sec}}} \mathbf{1}\{\arg\min_j d(\hat{e}_i, e_j) = k\}, \sigma^2)$ for $D_{\text{pri}}$. where

$$\left| \sum_{x_i \in D_{\text{sec}}} \mathbf{1}\{\arg\min_j d(\hat{e}_i, e_j) = k\} \right| \leq |D_{\text{sec}}|,$$

and $|D_{\text{sec}}|$ is distributed according to $\mu := \sum_{x_i \in D_{\text{sec}}} \text{Bern}(\rho_i)$. Hence, invoking Lemma 4.5 of Choquette-Choo et al. (2024), $\mathcal{N}(\mu, \sigma^2)$ and $\mathcal{N}(0, \sigma^2)$ forms a dominating pair that bounds the blow-up function for $P$ and $Q$. Similarly, for cluster center:

$$\left\| \frac{n_k \cdot e_k + \sum_{i \in C_k} \hat{e}_{\text{pri},i}}{n_k + m_k} - e_k \right\| = \left\| \frac{\sum_{i \in C_k} \hat{e}_{\text{pri},i} - m_k \cdot e_k}{n_k + m_k} \right\| = \left\| \frac{\sum_{i \in C_k} (\hat{e}_{\text{pri},i} - e_k)}{n_k + m_k} \right\|$$
$$\leq \frac{2R}{n_k + m_k} \cdot \mu \leq \frac{2R}{n_k} \cdot \mu$$

For each iteration, since the noise is calibrated to bound the blow-up function, Theorem 2 of (Hayes et al., 2023) directly implies that the mechanism satisfies $(p, r)$-secret protection. $\square$

