# OpenReview forum: "Secret-Protected Evolution for Differentially Private Synthetic Text Generation"
_ICLR.cc/2026/Conference — ICLR 2026 Poster_

### Official Review · Reviewer_2rqM · 2025-10-30

**Soundness:** 3
**Presentation:** 3
**Contribution:** 3
**Rating:** 6
**Confidence:** 3

**Summary:**

The paper proposes **Secret-Protected Evolution (SecPE)**, a novel framework for generating high-fidelity synthetic text under formal privacy constraints. Unlike traditional Differential Privacy (DP), which enforces uniform guarantees across all data points, SecPE introduces the concept of **Secret Protection**, focusing on safeguarding specific sensitive content (“secrets”) rather than entire records. The framework extends the Private Evolution (PE) paradigm and integrates a new **Secret Clustering** module, which leverages public data and limited noisy private updates to build secret-aware cluster centers for efficient selection. In the **Protected Evolution** phase, candidate synthetic texts are iteratively generated and selected based on similarity to these noisy representatives. Theoretical analysis shows that SecPE satisfies $(p, r)$-secret protection—a relaxation of Gaussian Differential Privacy (GDP)—and yields tighter utility–privacy trade-offs. Empirical results on **OpenReview**, **PubMed**, and **Yelp** datasets demonstrate that SecPE achieves lower FID, higher downstream accuracy, and significantly reduced computational cost compared to µ-GDP–based Aug-PE baselines.

**Strengths:**

### Strengths
1. The paper introduces a **new privacy notion, (p, r)-secret protection**, that generalizes and relaxes Gaussian DP, offering a theoretically grounded yet more flexible privacy-utility balance.
2. The proposed **SecPE algorithm** effectively combines secret-aware protection with the efficiency of K-means–based clustering, reducing computational complexity from $O(MN_{syn})$ to $O(KN_{syn})$.
3. **Empirical results** on multiple datasets show consistent performance improvements, with SecPE achieving higher downstream accuracy and lower FID values compared to Aug-PE.
4. The framework provides a **clear theoretical linkage** between secret protection and existing DP frameworks (DP, GDP), thereby establishing conceptual coherence.
5. The experimental section is thorough, including analyses on efficiency, downstream accuracy, FID, and ablations over LLM size and clustering hyperparameters.

### Quality
The paper is technically solid and theoretically well-founded. The connection between secret protection and Gaussian DP is clearly derived and mathematically consistent. The experimental evaluation is extensive and replicable. However, the lack of a real-world deployment or qualitative human evaluation of the generated text slightly limits the practical validation.

## Clarity
The paper is **clearly written and well structured**, especially in the methodology section. Figures and algorithms (notably Algorithms 1–3) provide a good overview of the pipeline. However, some mathematical symbols (e.g., blow-up function and trade-off function) could be briefly explained when first introduced to enhance readability for non-privacy experts.

### Significance
SecPE represents a **notable conceptual advance** in privacy-preserving text generation, shifting from record-level to secret-level guarantees. Its relaxation of GDP could inspire a new line of research in secret-aware privacy mechanisms, potentially extending beyond text to multimodal settings. The method’s computational efficiency also makes it practical for real-world applications where DP-finetuning is infeasible.

**Weaknesses:**

### Weaknesses
1. The paper would be strengthened by **a more comprehensive comparison with DP-finetuned LLMs**, which, while computationally costly, have been shown to perform competitively in private text synthesis.
2. Although the paper reports improvements in synthetic quality, **SecPE’s performance on personally identifiable information (PII)** tasks shows only marginal gains. The method’s real-world effectiveness in dense secret scenarios therefore remains somewhat inconclusive.
3. The definition of "secret" remains **qualitative and application-dependent**. The paper acknowledges this limitation but does not explore strategies for adaptive or data-driven secret detection beyond keyword-level identification.
4. The **sensitivity of the algorithm to different clustering sizes (K)** and to varying prior parameters $(p, r)$ could be analyzed more rigorously, as these hyperparameters directly influence both privacy guarantees and utility.

**Questions:**

1. How sensitive is SecPE’s performance to the choice of secret prior probabilities \(p_j\)? Would a mis-specified prior significantly degrade the protection or utility?
2. Could the authors discuss how SecPE scales when the number of secrets grows very large (e.g., thousands of potential sensitive attributes)?
3. The authors mention that clustering abstracts away fine-grained details. Could this abstraction lead to mode collapse or reduced diversity in the generated synthetic data?
4. In *user-level DP*, the protection unit is an **entire user’s data** (i.e., all records belonging to a user), whereas SecPE protects **individual secrets** regardless of user identity. Could SecPE be extended to user-level protection, or combined with it to achieve multi-granular privacy?

---

> ### Author Response · Authors · 2025-11-18
> **Response to reviewer 2rqM (Part 1)**
>
> We appreciate the reviewer’s insightful comments. Below we provide our responses and clarifications.
>
> $\textbf{Q1: Regarding the comparison with DP-finetuned LLMs}$
> 1. Previous studies such as (Xie et al., 2024; Zou et al., 2025; Hou et al., 2025) have shown that private evolution (PE) methods consistently outperform DP-based generators. This is because the clipping and noising operations in DP-SGD introduce significant distortion to the model updates. And finetuning requirement prevent effective use of state-of-the-art closed-source LLMs.
> 2. Moreover, the secret-protection framework should not be directly transferred to a generator; the algorithm must be specifically designed for this purpose. We plan to further improve such fine-tuning frameworks as part of our future work.
>
> $\textbf{Q2: PII experiment and dense secret scenarios.}$
> 1. We acknowledge that the improvement in the PII experiment reported in the paper is marginal. However, the performance gain over the original PE strongly depends on the quality of the PII detection tool. To further verify robustness, we conducted an additional PII experiment using a better detection tool from Microsoft (https://github.com/microsoft/presidio). The new results show a significant improvement, particularly under strong privacy settings (e.g., $r/p = 2$ or $10$), confirming that SecPE remains effective when reliable secret detection is available.
>
> |   Method  |   r/p=2  |        |  r/p=10  |        |  r/p=50  |        | r/p=\infty |        |
> |:---------:|:--------:|:------:|:--------:|:------:|:--------:|:------:|:----------:|:------:|
> |           | Category | Rating | Category | Rating | Category | Rating |  Category  | Rating |
> |   AugPE   |   71.72  |  55.80 |   72.54  |  63.34 |   74.15  |  66.92 |    75.14   |  67.04 |
> | SecPE_600 |   73.88  |  64.80 |   74.38  |  66.58 |   75.40  |  66.66 |    75.34   |  65.50 |
>
> 2. First, we would like to clarify what “dense” and “sparse” mean in our context. In both the PII and random-word frequency tasks, the private data are dense within the overall dataset. The key difference lies in the distribution of secrets: in the previous PII task (possibly due to misidentification by the detection tool), one specific secret appears far more frequently than others, dominating the overall protection. This situation effectively degenerates into the single-secret protection case. As illustrated in Appendix B, in such cases, the corresponding noise ratio is relatively small.
> 3. With the new PII detection tool, the private data become less dense, and no single secret dominates the dataset. In this case, the improvement is most significant.
>
> $\textbf{Q3: Secret detection beyond keyword-level identification}$
> 1. We agree that the concept of secret remains qualitative and application-dependent. However, we believe this subjectivity is essential rather than a limitation. The definition of a secret should be determined by the data provider, who decides what information needs protection. For example, a hospital may wish to protect patient information, a pharmaceutical company may protect drug formulas, and a financial institution may protect trading data.
> 2. Once the secrets are defined, the detection process yields a binary matrix, where each row vector represents the secrets contained in one data point, this matrix serves as the actual input to our method. Regarding adaptive or data-driven secret detection, we view this as a promising future research direction, analogous to the development of PII detection techniques.
>
> $\textbf{Q4: Sensitivity of the algorithm to different clustering sizes}$
> 1. You are absolutely right about the mode collapse issue in the generated synthetic data. However, this phenomenon only occurs when $K$ is too small. We discuss this hyperparameter in Lines 241–244, where we state that the number of clusters $K$ should scale with (i) the size of the original dataset and (ii) the target number of synthetic samples. Empirically, the performance is quite insensitive to the exact choice of $K$, as long as $K$ is not too small, where it fails to capture the full data information.
>
> | epoch 20 | K=50  | K=100 | K=200 | K=800 | K=1200 | K=1600 |
> |----------|-------|-------|-------|-------|--------|--------|
> | Category | 62.50 | 63.88 | 70.94 | 73.76 | 73.56  | 73.92  |
> |  Rating  | 61.84 | 64.46 | 64.72 | 66.16 | 67.34  | 67.18  |
>
> 2. On the other hand, we intentionally avoid choosing $K$ on the same order as $N_{\mathrm{private}}$, since the extreme case of $K = N_{\mathrm{private}}$ degenerates to Aug-PE, which is not the goal of our framework.

---

> ### Author Response · Authors · 2025-11-18
> **Response to reviewer 2rqM (Part 2)**
>
> $\textbf{Q5: Mis-specified prior}$ $p_j$
> 1. Thank you for the question. The short answer is that a mis-specified prior does not cause significant degradation in performance. As discussed in Appendix C, where we establish the connection with $(\epsilon, \delta)$-DP, the ratio $r/p$ plays a role analogous to $\epsilon$ when $p$ is small. Here, $r/p$ represents the ratio between the posterior and prior probabilities, which quantifies the level of secret protection. Conceptually, we do not intend to amplify the prior, as the framework is designed to calibrate protection based on realistic prior knowledge rather than to artificially exaggerate it.
>
> $\textbf{Q6: Scales of SecPE when the number of secrets grows}$
> 1. The scale of the secrets does not have a large effect on SecPE, but the density of secrets does. More specifically, only the proportion of private data matters. In the extreme case where every sample is treated as containing a secret, the setting degenerates back to the single-secret case, where we can only guarantee smaller noise.
> 2. We want to emphasize that our focus is precisely on the most realistic scenarios, where the dataset may contain a large number of secrets, but the data containing those secrets are not dense. In such settings, SecPE provides meaningful and consistent improvement over the baseline PE framework.
>
> $\textbf{Q7: User-level DP problem}$
> 1. First, we would like to clarify that our definition focuses on data reconstruction protection, rather than membership inference.
> 2. Regarding the user-level setting, we can simply extract the secrets associated with each user and combine them into a unified secret set. More specifically, suppose user A has secrets {$s_a^1, \ldots, s_a^{n_a}$} and user B has secrets {$s_b^1, \ldots, s_b^{n_b}$}, where the $s_a$’s are distinct from the $s_b$’s. Then, the linear program (6) automatically computes the corresponding weights for each user independently, because the weights for user A’s data are computed under the constraint {${x_i \in D_{\text{pri}} \mid s_a \in x_i}$}, which does not affect user B.

---

### Official Review · Reviewer_Rcqb · 2025-10-31

**Soundness:** 3
**Presentation:** 2
**Contribution:** 3
**Rating:** 6
**Confidence:** 3

**Summary:**

This paper proposes Secret-Protected Evolution (SecPE), a novel framework for generating synthetic text that addresses key limitations in existing privacy-preserving methods. The authors identify that standard Differential Privacy (DP), imposes a uniform privacy guarantee that over-protects non-sensitive data, leading to unnecessary utility loss. The paper provides a strong theoretical foundation, proving the pipeline satisfies (p, r)-secret protection. Empirically, evaluations on OpenReview, PubMed, and Yelp benchmarks show SecPE consistently outperforms a GDP-based baseline.

**Strengths:**

Strengths include the novel and practical privacy formulation, some computational efficiency gains, comprehensive evaluation across multiple models and datasets

**Weaknesses:**

(1) The framework also relies on a clear definition of "secrets" and their prior probabilities (p), which is left somewhat ambiguous and could be a practical hurdle.

(2) The presentation needs some improvement.   For example, the sentence on LIne 75-77 is not understandable.   The definitions needs some background knowledge, which is absent.

(3) Finally, performance improvements are more modest when secrets are dense (as in a PII task), and the method's dependence on representative public data for clustering is not fully explored.

**Questions:**

(1) How can secrets be systematically defined and their priors quantified in real-world scenarios?

(2) Could adaptive clustering mitigate the minor utility loss in non-private settings?

(3) How does the framework compose over multiple data releases

---

> ### Author Response · Authors · 2025-11-18
> **Response to reviewer Rcqb**
>
> Many thanks for the informative review. Below, we provide our responses to the reviewer’s concerns.
>
> $\textbf{Q1: Clear definition of secrets}$
> 1. We want to clarify that the definition of secrets is inherently subjective. Personally identifiable information (PII) is one clear example, which also requires pre-definition and may evolve dynamically. In practical applications, a hospital may wish to protect patient information, while a financial institution may wish to protect trading records. In summary, the responsibility for defining what to protect lies with the data provider, rather than with our framework.
>
> $\textbf{Q2: Quantified of priors in real-world scenarios}$
> 1. In our definition, $p$ represents the attacker’s prior success probability. When a secret appears with non-uniform frequency in the data, a realistic attacker could exploit this information; thus, using the empirical frequency as $p$ is a conservative and practical choice. When the attacker has no side information, we adopt a random-guess prior—for example, a 4-digit numeric password has $10^4$ possibilities, so $p = 1/10^4$. Our guarantee is defined for any given $p$, allowing practitioners to select the value that reflects their specific threat model.
> 2. As noted in our response to Reviewer 2rqM, a mis-specified prior does not cause significant degradation in performance. As discussed in Appendix C, where we establish the connection with $(\epsilon, \delta)$-DP, the ratio $r/p$ plays a role analogous to $\epsilon$ when $p$ is small. Conceptually, we do not intend to amplify the prior. The ratio $r/p$ represents the relationship between the posterior and prior probabilities, which quantifies the level of secret protection.
>
> $\textbf{Q3: Absent background knowledge in line 75-77}$
> 1. Basically, we are saying that once the secrets are predefined, we can detect the corresponding information, and the non-sensitive (i.e., non-secret) data can be freely used for clustering and summarization. This allows the algorithm to operate more efficiently, since the privacy budget is applied only to the secret-related data.
> 2. We revised the main text (Line 76-79) to make it clearer and more understandable.
>
> $\textbf{Q4: Dependence on representative public data}$
> 1. Here we would like to clarify that once the secrets are defined, the division between private and public data becomes fixed. A good representation of the public data can significantly improve performance (see also our ablation study on $K$). However, if the representation fails to capture the full data structure, as in the case when $K$ is too small, the performance will degrade.
>
> | epoch 20 | K=50 | K=100 | K=200 | K=800 | K=1200 | K=1600 |
> |-------|-------|-------|-------|-------|--------|--------|
> | Category | 62.50 | 63.88 | 70.94 | 73.76 | 73.56 | 73.92 |
> | Rating | 61.84 | 64.46 | 64.72 | 66.16 | 67.34 | 67.18 |
>
> $\textbf{Q5: Concers on PII experiment}$
> 1. We would like to clarify that the performance gain over the original PE strongly depends on the quality of the PII detection tool. To further verify robustness, we conducted an additional PII experiment using a better detection tool from Microsoft (https://github.com/microsoft/presidio). The new results show a significant improvement, particularly under strong privacy settings (e.g., $r/p = 2$ or $10$), confirming that SecPE remains effective when reliable secret detection is available.
> | Method | r/p=2 | | r/p=10 | | r/p=50 | | r/p=\infty | |
> |:---------:|:--------:|:------:|:--------:|:------:|:--------:|:------:|:----------:|:------:|
> | | Category | Rating | Category | Rating | Category | Rating | Category | Rating |
> | AugPE | 71.72 | 55.80 | 72.54 | 63.34 | 74.15 | 66.92 | 75.14 | 67.04 |
> | SecPE_600 | 73.88 | 64.80 | 74.38 | 66.58 | 75.40 | 66.66 | 75.34 | 65.50 |
>
> $\textbf{Q6: Could adaptive clustering mitigate the minor utility loss in non-private settings?}$
> 1. We appreciate this insightful suggestion and agree that adaptive clustering could further improve performance. In this work, our main focus is to examine whether representing the entire dataset through clustering is feasible, and we adopted K-means as the most straightforward approach to demonstrate this. We believe that more advanced or adaptive representation methods could yield even better results, and we plan to explore this direction in future work.
>
> $\textbf{Q7: How does the framework compose over multiple data releases}$
> 1. We assume the question refers to the multi-source data setting (please correct us if this interpretation is not intended). In this case, our framework can be naturally extended by performing clustering independently for each source and then taking the union of all clusters, i.e., the total number of clusters becomes $K = \sum_s K_s$.
>
> We hope these clarifications address your concerns.

---

### Official Review · Reviewer_ntT2 · 2025-10-31

**Soundness:** 4
**Presentation:** 3
**Contribution:** 3
**Rating:** 4
**Confidence:** 4

**Summary:**

This paper presents a novel Secret Protection Evolutionary framework (SecPE) for differentially private (DP) synthetic text generation. The work aims to address the issue of significant utility loss and high computational overhead associated with traditional DP methods in text generation. By introducing the concept of Secret-Protected Evolution	as an alternative to uniform DP guarantees,  it achieves a better utility-privacy trade-off. In the theoretical part, the paper defines a (p, r)-secret protection criterion and establishes its connection to Gaussian DP. The experimental section validates the advantages of SecPE across several benchmark datasets (OpenReview, PubMed, Yelp), demonstrating reductions in Fréchet Inception Distance (FID), improved accuracy on downstream tasks, and decreased runtime.

**Strengths:**

1.This work proposes a (p,r)-secret protection framework and establishes its theoretical connection to Gaussian Differential Privacy (GDP), offering a new perspective for privacy-preserving research.
2.The experimental design is comprehensive, and the validation is thorough.

**Weaknesses:**

1.The paper operates on the assumption that "secrets" can be predefined; however, in real-world scenarios, sensitive content is often dynamic and non-enumerable.
2.The paper fails to analyze the impact of the cluster number K on the effectiveness (it only mentions "insensitive" but provides no ablation studies).
3.The improvement on the PII task is relatively limited (Table 7), potentially due to high secret density diminishing the protection benefits. Further analysis on the impact of secret sparsity is required.

**Questions:**

The manuscript employs the term "operative point" on multiple occasions without providing a formal definition.

---

> ### Author Response · Authors · 2025-11-18
> **Response to reviewer ntT2**
>
> We appreciate your thoughtful comments and constructive suggestions, which have greatly helped us improve the paper. And we have carefully addressed all comments below.
>
> $\textbf{Q1: Sensitive content is often dynamic and non-enumerable.}$
> 1. First, we would like to clarify that the definition of secrets is inherently subjective. Personal identifiable information (PII) is just one natural example of a secret, which also requires pre-definition and may evolve dynamically.
> 2. Our approach is designed for the data provider, who determines which information should be protected. For example, a hospital may wish to protect patient information, a pharmaceutical company may protect drug formulas, and a financial institution may protect trading data.
> 3. The key contribution and focus of our work lie in providing a mechanism to protect these defined secrets, which prior methods either fail to achieve or overprotect, as they are not capable of operating at the secret level.
>
> $\textbf{Q2: Impact of the cluster number K}$
> 1. This concern was also raised by other reviewers, and we appreciate the opportunity to clarify it.
> The mode collapse issue in the generated synthetic data occurs only when $K$ is too small. We conducted an ablation study on different values of $K$, and the results show that the performance is largely insensitive to the exact choice of $K$, as long as $K$ is not too small, where it fails to capture the full data information.
>
> | epoch 20 | K=50 | K=100 | K=200 | K=800 | K=1200 | K=1600 |
> |-------|-------|-------|-------|-------|--------|--------|
> | Category | 62.50 | 63.88 | 70.94 | 73.76 | 73.56 | 73.92 |
> | Rating | 61.84 | 64.46 | 64.72 | 66.16 | 67.34 | 67.18 |
>
> 2. We hope this explanation sufficiently addresses your concern and clarifies our design choice.
>
> $\textbf{Q3: The improvement on the PII task is relatively limited}$
> 1. We acknowledge that the improvement in the PII experiment reported in the paper is marginal. However, the performance gain over the original PE strongly depends on the quality of the PII detection tool. To further verify robustness, we conducted an additional PII experiment using a better detection tool from Microsoft (https://github.com/microsoft/presidio). The new results show a significant improvement, particularly under strong privacy settings (e.g., $r/p = 2$ or $10$), confirming that SecPE remains effective when reliable secret detection is available.
>
> | Method | r/p=2 | | r/p=10 | | r/p=50 | | r/p=$\infty$ | |
> |:---------:|:--------:|:------:|:--------:|:------:|:--------:|:------:|:----------:|:------:|
> | | Category | Rating | Category | Rating | Category | Rating | Category | Rating |
> | AugPE | 71.72 | 55.80 | 72.54 | 63.34 | 74.15 | 66.92 | 75.14 | 67.04 |
> | SecPE_600 | 73.88 | 64.80 | 74.38 | 66.58 | 75.40 | 66.66 | 75.34 | 65.50 |
>
> 2. You are absolutely right that the sparsity of secrets affects the performance. More specifically, the density of secrets determines the proportion of private data. In the extreme case where every sample is treated as containing a secret, the setting degenerates back to the single-secret case, where we can only guarantee smaller noise. The numerical simulation in Appendix B further supports this observation, showing that sparser secrets lead to greater improvement. The assumption that secret-containing data are sparse within the dataset is precisely the motivation of our proposed framework.
>
> $\textbf{Q4: Operative point problem}$
> 1. We apologize for the unclear term, and we have corrected this in the main text (Line 81-83). This phrase appears only once in the paper (Line 80). In the context of Gaussian DP, the privacy notion requires that the entire distribution trade-off curve dominates the Gaussian trade-off. The operative point we refer to here indicates that, in our setting, we only need to protect a specific point $(p_j, r_j)$ on this curve, rather than the full trade-off curve. Hence, this can be viewed as a relaxation of Gaussian DP.

---

> ### Author Response · Authors · 2025-11-26
>
> Dear Reviewer,
>
> I hope this message finds you well. Thank you once again for your thoughtful effort and valuable comments. We have carefully addressed all the main concerns in detail, and we hope that our responses are satisfactory. As the discussion phase is nearing its end, we warmly welcome any additional suggestions or questions you may have. We would be very happy to further clarify any remaining concerns.
>
> Best, Authors

---

### Official Review · Reviewer_Gvyt · 2025-11-14

**Soundness:** 4
**Presentation:** 2
**Contribution:** 2
**Rating:** 6
**Confidence:** 3

**Summary:**

The authors adapt private evolution to offer guarantees under the “secret-protection” framework, which is a recent DP relaxation proposed by (Ganesh et al., 2025). They introduce a clustering step with public data to improve efficiency of PE. They conduct experiments by instantiating the secret-protection framework by declaring random words as “secrets” and examples containing them to betray the secret, showing utility improvements over regular PE that provides uniform protection.

**Strengths:**

- The topic is of high importance to the community. Secret protection is a promising DP relaxation that can potentially address many “type mismatches” between DP and the kind of sensitive data that companies own and want to extract value out of. Synthetic data via private evolution is one of the least resource- and expertise-intensive ways to get something out of your sensitive data. For these two reasons, finding a good design here will unlock a lot of useful practical applications.

- The experiments demonstrating utility improvement of SecPE compared to PE, in the setting where the secret protection definition is instantiated by declaring PII or random words as “secrets” (with examples containing them considered to betray the secret), are well-executed and highly interesting.

**Weaknesses:**

- As secret protection is a relatively nascent privacy framework, I believe the authors could do more in terms of exposition in Section 3.1, as well as (1) give the specific instantiation of (S,E,T) (in the notation of (Ganesh et al., 2025)) that defines the neighboring relationship fundamental to applications of secret protection, and (2) describe how secret clustering relates to secret protection.

  - Specifically, the authors give a relatively terse statement of the secret-protection definition. Lemma 3.2 proves GDP implies a particular instantiation of secret protection. Rather than focusing on this point, the main text would be better served to describe further the algorithmic changes to PE enabled by relaxation of the DP definition.

- The core algorithmic tool for enabling the secret protection guarantee is from (Ganesh et al., 2025). Nothing particular PE under secret protection is introduced.

- While authors report computational efficiency gains from SecPE, these gains feel misattributed. Reported speedup is in terms of improving pairwise similarity calculations. First, it is relatively unintuitive to me that the bottleneck is similarity computation, rather than iterative rewrites with a possibly expensive API model. Perhaps a FLOPs or memory analysis of the operation, compared to inference FLOPs and memory would shed some light here. Second, avoiding pairwise similarity computations seems orthogonal to SecPE, and there exist many simple and practical solutions to the problem for regular PE. For example:
  - Using approximate nearest neighbours search over the index of synthetic examples (https://github.com/google-research/google-research/tree/master/scann).
  - Clustering the synthetic data, and voting on clusters. Although not used in PE, this is employed in the highly-related work on private postprocessing (https://arxiv.org/abs/2402.13659), which can be viewed as half of one step of private evolution.

**Questions:**

- Secret protection is instantated by a (S,E,T) tuple (in the notation of Ganesh et al, 2025). In experiments, what is T?

- Any intuitions about how the number of secrets and their choice affect the utility improvement of SecPE compared to PE? Is this entirely predicted by the computed noise and sampling parameters?

---

> ### Author Response · Authors · 2025-11-18
> **Response to reviewer Gvyt (Part 1)**
>
> We sincerely thank the reviewer for the thoughtful and constructive feedback. We address the comments and provide clarifications below
> $\textbf{Q1: Specific instantiation of (S,E,T) and describe how secret clustering relates to secret protection.}$
>
> Thanks for bringing up the question about the definition. Let us clarify the precise meaning.
> 1. Given a dataset {$\{x_1, ..., x_n\}$} and a secret set S ={${s_1, ..., s_m}$}, In the original paper, the define the function $E$ maps each $x_i$ to a ${0,1}^m$ vector, indicating whether $x_i$ contains secret $s_j$. The inclusion of $E$ means that we assume knowledge of the secret distribution within the dataset, allowing secret detection to be applied beforehand. The function $T_j$ is an indicator that determines whether $x_i$ and $x_i^{\prime}$ differ only in secret $s_j$. Together with $E$, for each secret $s_j$, they jointly define the neighboring relation between two datasets and how the adversarial dataset is constructed.
> 2. We simplify the definition and incorporate it into the main text (Line 161-167) to better illustrate the neighboring relationship and enhance overall readability. However, since the protection guarantee is defined in terms of the best prior attack probability $p_j$, the notations $E$ and $T$ are no longer needed beyond this point. In experiments, as long as we can evaluate $p_j$, the specific form of $T$ becomes unnecessary.
> 3. Regarding the secret clustering algorithm, we provide a rigorous proof (Theorem~1) showing that it satisfies the the secret clustering algorithm satisfies $(p,r)$-secret protection definition.
>
> $\textbf{Q2: Describe further the algorithmic changes to PE enabled by relaxation of the DP definition}$
> 1. In the secret-protection definition, we only need to protect the predefined secrets rather than the entire dataset. Compared with the DP-PE algorithm, our approach first fully utilizes the non-private data for clustering, which is the first modification to the original PE algorithm. Second, the linear program computes a sampling probability for each secret-containing (private) data point and then adjusts the cluster centers with noise that is strictly smaller than the DP noise. After this step, the evolution system performs voting based on the adjusted centers, rather than repeatedly accessing the entire dataset. One important note is that, for a fair comparison, we did not modify the public-data voting histogram in the DP baseline; that is, noise is applied only to the votes of private data.
>
> $\textbf{Q3: Improvement over PE under secret protection}$
> 1. First, the original secret-protection definition in (Ganesh et al., 2025) was developed for the SGD algorithm, where the noise calibration is based on the KL-divergence formulation. We adopt this definition but extend it to the PE framework by introducing a trade-off function that aligns with the Gaussian DP interpretation.
> 2. As stated in Section 3.2.2, “in the PE procedure, a small number of synthetic samples receive the majority of votes”, we improve the efficiency of PE by incorporating a clustering step. Moreover, as “predefined secrets allow us to first detect and cluster using only public data”, we propose a secret clustering algorithm that further enhances the original PE procedure under the secret-protection setting.

---

> ### Author Response · Authors · 2025-11-18
> **Response to reviewer Gvyt (Part 2)**
>
> $\textbf{Q4: Efficiency bottleneck and API question}$
> 1.  As shown in Table 2, the histogram computation in one epoch of the original PE algorithm on the Yelp dataset requires around 8 hours, whereas our method reduces this time to almost zero, since the large-scale dataset no longer needs to be processed after clustering.
>
> |            | Yelp  |           |
> |------------|-------|-----------|
> | Time (sec) | LLM   | Histogram |
> | AugPE      | 347.1 | 30126.4   |
> | SecPE      | 347.6 | 2.3       |
>
> 2. Regarding the use of expensive APIs, we acknowledge that the LLM processing time remains similar to the original method; however, this step is essential to ensure that the generated synthetic data accurately reflects the original data distribution. As long as we operate within the PE framework, such iterative generation is inevitable.  In practical deployment, we suggest using local open-source models with more training iterations instead of closed-source, expensive APIs, since our approach runs each iteration extremely efficiently. Below, we provide a result using a smaller local model with 20 epochs, which even outperforms the large API-based models.
>
> |                 | r/p = 10 |       | r/p = $\infty$ |       |
> |-----------------|----------|-------|--------------|-------|
> | GPT2 (Epoch 5)  | 73.82    | 58.36 | 72.96        | 62.22 |
> | Qwen-2.5-7B     | 74.56    | 63.06 | 74.24        | 63.34 |
> | GPT-4o-Mini     | 74.84    | 62.96 | 75.10        | 63.28 |
> | GPT2 (Epoch 20) |   74.50  | 66.46 |     75.50    | 66.24 |
>
> 3. Memory usage is significantly reduced in our approach because clustering is performed before the PE step. Specifically, we save about 25.1 GB of memory by avoiding the need to store embeddings for the entire dataset.Regarding GPU utilization, the original PE method achieves only 3.2%, whereas our approach reaches approximately 38.6%.
> 4. While approximate nearest-neighbor methods can reduce computational complexity, they inevitably lose fine-grained information, much like clustering. That said, we appreciate the reviewer’s comment on memory efficiency. This perspective is indeed very important and our clustering-based approach achieves substantial memory savings. As for the latter paper (Yu et al., 2024), it first trains a DP-generator, which is extremely costly and precisely the issue that PE is designed to avoid. Moreover, it performs clustering directly on private data, which we believe is unreasonable, as clustering on private data would undermine the diversity of the generated synthetic data. Besides, the complexity remains $\mathcal{O}(N_{\mathrm{pri}})$ (for Yelp, about 1.9 million), while our method reduces it to $\mathcal{O}(K)$.

---

### Author Response · Authors · 2025-11-29
**General response**

Dear Reviewers and Area Chair,

We are sincerely grateful for the substantial time and meticulous effort you have dedicated to handling our submission. We understand that going through the entire discussion can be quite time-consuming, so in order to save you some effort, we have prepared a brief summary of our rebuttal.

Our work proposes Secret-Protected Evolution (SecPE), a framework for generating synthetic text while **protecting specific sensitive content (“secrets”)**. Instead of enforcing uniform DP guarantees on all data points, which often causes large utility loss and high computational cost, SecPE focuses protection on these declared secrets. SecPE extends Private Evolution with a **Secret Clustering** module that uses public data and a few noisy private updates to guide generation. On several benchmark datasets, SecPE produces synthetic data with **higher fidelity to the original data**, with **better downstream accuracy** and **lower computational cost** than DP-based baselines.

Among the four reviews, three are clearly positive, and only one reviewer gives a negative recommendation. For this more critical review, we have provided detailed responses and corresponding revisions in the rebuttal and revised manuscript, and we believe that all of the reviewer’s concerns have been fully resolved.

During this period, we are pleased to have received several positive remarks, including:

- The topic and problem setting are of high practical importance to the community. (Reviewer Gvyt)
- The paper proposes a novel (p, r)-secret protection notion that generalizes and relaxes Gaussian DP, with a clear theoretical linkage to existing DP/GDP frameworks. (Reviewers ntT2, 2rqM)
- SecPE provides computational efficiency gains by operating at the cluster level instead of over all private records. (Reviewers Rcqb, 2rqM)
- The experimental evaluation is comprehensive and well-designed, covering multiple models, datasets, and ablations. (Reviewers ntT2, Rcqb, 2rqM)
- The utility improvements of SecPE over PE, especially in the PII and random-word secret instantiations, are well-executed and interesting. (Reviewer Gvyt)

Based on the reviewers’ feedback, we have also improved the manuscript, especially in previously unclear parts and in experiments on specific points. The main additions and clarifications are summarized as follows:

- **Ablation analysis on the number of clusters \(K\).** We added a more detailed ablation study on different choices of \(K\), now included in **Section 4.2 and Table 6.** In brief, a good representation of the public data can substantially improve downstream performance, and empirically the method is quite insensitive to the exact choice of  \(K\) as long as  \(K\) is not too small. Mode collapse in the generated synthetic data tends to occur only when  \(K\) is too small to capture the full data distribution.
- **Additional PII experiment with a stronger detection tool.** We conducted a new experiment on Personally Identifiable Information (PII) using Microsoft Presidio as a stronger detection tool. The results are reported in **Appendix B.5 and Table 9**. Briefly, the performance gain of SecPE over the original PE is highly dependent on the quality of the PII detection tool. With the improved detector, the private data become less “dense” in the sense that no single secret overwhelmingly dominates the dataset. In this more genuinely multi-secret setting, the advantage of SecPE is most pronounced, which is consistent with the design goal of our method.
- **Memory and GPU utilization.** We reported the memory savings and GPU utilization for the Yelp experiment **(Lines 406–410)**, showing substantial gains from clustering before the PE step. Our secret-clustering design before PE yields substantial savings in both memory and runtime, and also significantly improves GPU utilization.
- **Clarification of neighboring dataset relationships and terminology.** We added a more detailed explanation of the neighboring dataset relationship in **Section 3.1** and revised the explanation of several unclear terms for better clarity in **Lines 72–84**.
- All modifications are highlighted in ${\color{blue}\text{blue}}$ in the revised main manuscript.

Thank you once again for your professional guidance and for your commitment to enhancing the quality of our work.

Best regards,

Authors of Submission 4301

---

### Meta-Review · Area_Chair_FfFp · 2026-01-05

**Summary:**

The submission received generally positive feedback, with reviewers praising the new theoretical grounding and the computational gains achieved through secret-aware clustering. While several initial concerns were raised, the authors provide new results during the rebuttal to address most of them.

**Reviewer Concerns:**

Here are the summary of concerns raised by Reviewer ntT2.

Addressed:

1. The paper fails to analyze the impact of the cluster number K on the effectiveness.

2. The improvement on the PII task is relatively limited.

Outstanding:

1. The paper operates on the assumption that "secrets" can be predefined; however, in real-world scenarios, sensitive content is often dynamic and non-enumerable.

**Reviewer Scores:**

6/6/6/6

---

### Decision · Program_Chairs · 2026-01-26

Accept (Poster)